# ON IMPROVING ADVERSARIAL TRANSFERABILITY OF VISION TRANSFORMERS

**Muzammal Naseer**[§†]    **Kanchana Ranasinghe**[∘]    **Salman Khan**[§]
**Fahad Shahbaz Khan**[§‡]    **Fatih Porikli**[▽]
[†]Australian National University,    [∘]Stony Brook University,    [‡]Linköping University
[§]Mohamed bin Zayed University of AI,    [▽]Qualcomm USA
`muzammal.naseer@anu.edu.au`

## ABSTRACT

Vision transformers (ViTs) process input images as sequences of patches via self-attention; a radically different architecture than convolutional neural networks (CNNs). This makes it interesting to study the adversarial feature space of ViT models and their transferability. In particular, we observe that adversarial patterns found via conventional adversarial attacks show very *low* black-box transferability even for large ViT models. We show that this phenomenon is only due to the sub-optimal attack procedures that do not leverage the true representation potential of ViTs. A deep ViT is composed of multiple blocks, with a consistent architecture comprising of self-attention and feed-forward layers, where each block is capable of independently producing a class token. Formulating an attack using only the last class token (conventional approach) does not directly leverage the discriminative information stored in the earlier tokens, leading to poor adversarial transferability of ViTs. Using the compositional nature of ViT models, we enhance transferability of existing attacks by introducing two novel strategies specific to the architecture of ViT models. *(i) Self-Ensemble:* We propose a method to find multiple discriminative pathways by dissecting a single ViT model into an ensemble of networks. This allows explicitly utilizing class-specific information at each ViT block. *(ii) Token Refinement:* We then propose to refine the tokens to further enhance the discriminative capacity at each block of ViT. Our token refinement systematically combines the class tokens with structural information preserved within the patch tokens. An adversarial attack when applied to such refined tokens within the ensemble of classifiers found in a single vision transformer has significantly higher transferability and thereby brings out the true generalization potential of the ViT's adversarial space. Code: `https://t.ly/hBbW`.

## 1 INTRODUCTION

Transformers compose a family of neural network architectures based on the self-attention mechanism, originally applied in natural language processing tasks achieving state-of-the-art performance (Vaswani et al., 2017; Devlin et al., 2018; Brown et al., 2020). The transformer design has been subsequently adopted for vision tasks (Dosovitskiy et al., 2020), giving rise to a number of successful vision transformer (ViT) models (Touvron et al., 2020; Yuan et al., 2021; Khan et al., 2021). Due to the lack of explicit inductive biases in their design, ViTs are inherently different from convolutional neural networks (CNNs) that encode biases e.g., spatial connectivity and translation equivariance. ViTs process an image as a sequence of patches which are refined through a series of self-attention mechanisms (transformer blocks), allowing the network to learn relationships between any individual parts of the input image. Such processing allows wide receptive fields which can model global context as opposed to the limited receptive fields of CNNs. These significant differences between ViTs and CNNs give rise to a range of intriguing characteristics unique to ViTs (Caron et al., 2021; Tuli et al., 2021; Mao et al., 2021; Paul & Chen, 2021; Naseer et al., 2021b).

Adversarial attacks pose a major hindrance to the successful deployment of deep neural networks in real-world applications. Recent success of ViTs means that adversarial properties of ViT models

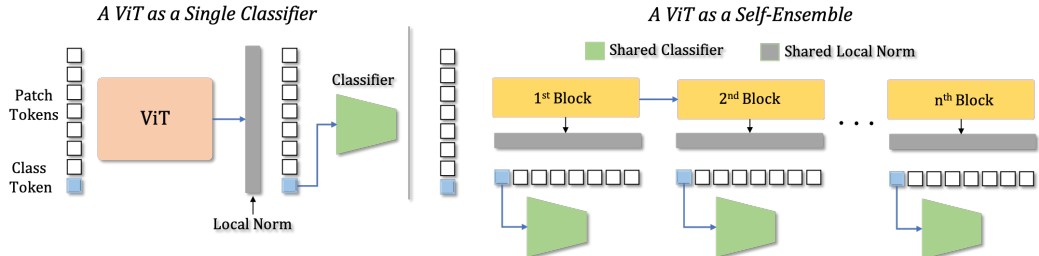

Figure 1: *Left:* Conventional adversarial attacks view ViT as a single classifier and maximize the prediction loss (e.g., cross entropy) to fool the model based on the last classification token only. This leads to sub-optimal results as class tokens in previous ViT blocks only indirectly influence adversarial perturbations. In contrast, our approach (*right*) effectively utilizes the underlying ViT architecture to create a self-ensemble using class tokens produced by all blocks within ViT to design the adversarial attack. Our self-ensemble enables to use hierarchical discriminative information learned by all class tokens. Consequently, an attack based on our self-ensemble generates transferable adversaries that generalize well across different model types and vision tasks.

also become an important research topic. A few recent works explore adversarial robustness of ViTs (Shao et al., 2021; Mahmood et al., 2021; Bhojanapalli et al., 2021) in different attack settings. Surprisingly, these works show that large ViT models exhibit lower transferability in black-box attack setting, despite their higher parameter capacity, stronger performance on clean images, and better generalization (Shao et al., 2021; Mahmood et al., 2021). This finding seems to indicate that as ViT performance improves, its adversarial feature space gets weaker. In this work, we investigate whether the weak transferability of adversarial patterns from high-performing ViT models, as reported in recent works (Shao et al., 2021; Mahmood et al., 2021; Bhojanapalli et al., 2021), is a result of weak features or a weak attack. To this end, we introduce a highly transferable attack approach that augments the current adversarial attacks and increase their transferability from ViTs to the unknown models. Our proposed transferable attack leverages two key concepts, multiple discriminative pathways and token refinement, which exploit unique characteristics of ViT models.

Our approach is motivated by the modular nature of ViTs (Touvron et al., 2020; Yuan et al., 2021; Mao et al., 2021): they process a sequence of input image patches repeatedly using multiple multi-headed self-attention layers (transformer blocks) (Vaswani et al., 2017). We refer to the representation of patches at each transformer block as *patch tokens*. An additional randomly initialized vector (*class token*[1]) is also appended to the set of patch tokens along the network depth to distill discriminative information across patches. The collective set of tokens is passed through the multiple transformer blocks followed by passing of the class token through a linear classifier (head) which is used to make the final prediction. The class token interacts with the patch tokens within each block and is trained gradually across the blocks until it is finally utilized by the linear classifier head to obtain class-specific logit values. The class token can be viewed as extracting information useful for the final prediction from the set of patch tokens at each block. Given the role of the class token in ViT models, we observe that class tokens can be extracted from the output of each block and each such token can be used to obtain a class-specific logit output using the final classifier of a pretrained model. This leads us to the proposed *self-ensemble* of models within a single transformer (Fig. 1). We show that attacking such a self-ensemble (Sec. 3) containing multiple discriminative pathways significantly improves adversarial transferability from ViT models, and in particular from the large ViTs.

Going one step further, we study if the class information extracted from different intermediate ViT blocks (of the self-ensemble) can be enhanced to improve adversarial transferability. To this end, we introduce a novel *token refinement* module directed at enhancing these multiple discriminative pathways. The token refinement module strives to refine the information contained in the output of each transformer block (within a single ViT model) and aligns the class tokens produced by the intermediate blocks with the final classifier in order to maximize the discriminative power of intermediate blocks. Our token refinement exploits the structural information stored in the patch tokens and fuses it with the class token to maximize the discriminative performance of each block. Both the *refined tokens* and *self-ensemble* ideas are combined to design an adversarial attack that is shown to significantly boost the transferability of adversarial examples, thereby bringing out the true

---

[1]Average of patch tokens can serve as a class token in our approach for ViT designs that do not use an explicit class token such as Swin transformer (Liu et al., 2021) or MLP-Mixer (Tolstikhin et al., 2021)

generalization of ViTs' adversarial space. Through our extensive experimentation, we empirically demonstrate favorable transfer rates across different model families (convolutional and transformer) as well as different vision tasks (classification, detection and segmentation).

## 2  BACKGROUND AND RELATED WORK

**Adversarial Attack Modeling:** Adversarial attack methods can be broadly categorized into two categories, *white-box* attacks and *black-box* attacks. While the white-box attack setting provides the attacker full access to the parameters of the target model, the black-box setting prevents the attacker from accessing the target model and is therefore a harder setting to study adversarial transferability.

**White-box Attack:** Fast Gradient Sign Method (FGSM) (Goodfellow et al., 2014) and Projected Gradient Descent (PGD) (Madry et al., 2018) are two initially proposed white-box attack methods. FGSM corrupts the clean image sample by taking a single step within a small distance (perturbation budget $\epsilon$) along the objective function's gradient direction. PGD corrupts the clean sample for multiple steps with a smaller step size, projecting the generated adversarial example onto the $\epsilon$-sphere around the clean sample after each step. Other state-of-the-art white-box attack methods include Jacobian-based saliency map attack (Papernot et al., 2016), Sparse attack (Modas et al., 2019), One-pixel attack (Su et al., 2019), Carlini and Wagner optimization (Carlini & Wagner, 2017), Elastic-net (Chen et al., 2018), Diversified sampling (Tashiro et al., 2020), and more recently Auto-attack (Croce & Hein, 2020b). We apply white-box attacks on surrogate models to find perturbations that are then transferred to black-box target models.

**Black-box Attack and Transferability:** Black-box attacks generally involve attacking a source model to craft adversarial signals which are then applied on the target models. While gradient estimation methods that estimate the gradients of the target model using black-box optimization methods such as Finite Differences (FD) (Chen et al., 2017; Bhagoji et al., 2018) or Natural Evolution Strategies (NES) (Ilyas et al., 2018; Jiang et al., 2019) exist, these methods are dependent on multiple queries to the target model which is not practical in most real-world scenarios. In the case of adversarial signal generation using source models, it is possible to directly adopt white-box methods. In our work, we adopt FGSM and PGD in such a manner. Methods like (Dong et al., 2018) incorporate a momentum term into the gradient to boost the transferability of existing white-box attacks, building attacks named MIM. In similar spirit, different directions are explored in literature to boost transferability of adversarial examples; *a) Enhanced Momentum:* Lin *et al.* (Lin et al., 2019) and Wang *et al.* (Wang & He, 2021) improve momentum by using Nesterov momentum and variance tuning respectively during attack iterations, *b) Augmentations:* Xie *et al.* (Xie et al., 2019) showed that applying differentiable stochastic transformations can bring diversity to the gradients and improve transferability of the existing attacks, *c) Exploiting Features:* Multiple suggestions are proposed in the literature to leverage the feature space for adversarial attack as well. For example, Zhou *et al.* (Zhou et al., 2018) incorporate the feature distortion loss during optimization. Similarly, (Inkawhich et al., 2020b;a; Huang et al., 2019) also exploit intermediate layers to enhance transferability. However, combining the intermediate feature response with final classification loss is non-trivial as it might require optimization to find the best performing layers (Inkawhich et al., 2020b;a), and *d) Generative Approach:* Orthogonal to iterative attacks, generative methods (Poursaeed et al., 2018; Naseer et al., 2019; 2021a) train an autoencoder against the white-box model. In particular, Naseer *et al.* show that transferability of an adversarial generator can be increased with relativistic cross-entropy (Naseer et al., 2019) and augmentations (Naseer et al., 2021a). Ours is the first work to address limited transferability of ViT models.

**The Role of Network Architecture:** Recent works exploit architectural characteristics of networks to improve the transferability of attacks. While Wu et al. (2020) exploit skip connections of models like ResNets and DenseNets to improve black-box attacks, Guo et al. (2020) build on similar ideas focused on the linearity of models.

*Our work similarly focuses on unique architectural characteristics of ViT models to generate more transferable adversarial perturbations with the existing white-box attacks.*

**Robustness of ViTs:** Adversarial attacks on ViT models are relatively unexplored. Shao et al. (2021) and Bhojanapalli et al. (2021) investigate adversarial attacks and robustness of ViT models studying various white-box and black-box attack techniques. The transferability of perturbations from ViT models is thoroughly explored in (Mahmood et al., 2021) and they conclude that ViT

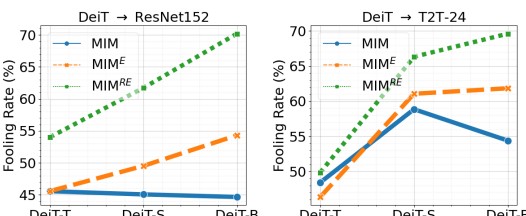

Figure 2: Adversarial examples for ViTs have only moderate transferability. In fact transferabililty (%) of MIM (Dong et al., 2018) perturbations to target models goes down as the source model size increases such as from DeiT-T (Touvron et al., 2020) (5M parameters) to DeiT-B (Touvron et al., 2020) (86M parameters). However, the performance of the attack improves significantly when applied on our proposed ensemble of classifiers found within a ViT (MIM$^E$ & MIM$^{RE}$).

models do not transfer well to CNNs, whereas we propose a methodology to solve this shortcoming. Moreover, Mahmood et al. (2021) explores the idea of an ensemble of CNN and ViT models to improve the transferability of attacks. Our proposed ensemble approach explores a different direction by converting a single ViT model into a collection of models (self-ensemble) to improve attack transferability. In essence, our proposed method can be integrated with existing attack approaches to take full advantage of the ViTs' learned features and generate transferable adversaries.

## 3 ENHANCING ADVERSARIAL TRANSFERABILITY OF VITS

**Preliminaries:** Given a clean input image sample $x$ with a label $y$, a source ViT model $\mathcal{F}$ and a target model $\mathcal{M}$ which is under-attack, the goal of an adversarial attack is generating an adversarial signal, $x'$, using the information encoded within $\mathcal{F}$, which can potentially change the target network's prediction ($\mathcal{M}(x')_{argmax} \neq y$). A set of boundary conditions are also imposed on the adversarial signal to control the level of distortion in relation to the original sample, i.e., $\|x - x'\|_p < \epsilon$, for a small perturbation budget $\epsilon$ and a $p$-norm, often set to infinity norm ($\ell_\infty$).

**Motivation:** The recent findings (Shao et al., 2021; Mahmood et al., 2021) demonstrate low black-box transferability of ViTs despite their higher parametric complexity and better feature generalization. Motivated by this behaviour, we set-up a simple experiment of our own to study the adversarial transferability of ViTs (see Fig. 2). We note that transferability of adversarial examples found via momentum iterative fast gradient sign method (Dong et al., 2018) (MIM) at $\ell_\infty \leq 16$ on DeiT (Touvron et al., 2020) does not increase with model capacity. In fact, adversarial transferability from DeiT base model (DeiT-B) on ResNet152 and large vision transformer (ViT-L (Dosovitskiy et al., 2020)) is lower than DeiT tiny model (DeiT-T). This is besides the fact that DeiT-B has richer representations and around $17\times$ more parameters than DeiT-T. We investigate if this behavior is inherent to ViTs or merely due to a sub-optimal attack mechanism. To this end, we exploit unique architectural characteristics of ViTs to first find an ensemble of networks within a single pretrained ViT model (self-ensemble, right Fig. 1). The class token produced by each self-attention block is processed by the final local norm and classification MLP-head to refine class-specific information (Fig. 2). In other words, our MIM$^E$ and MIM$^{RE}$ variants attack class information stored in the class tokens produced by *all* the self-attention blocks within the model and optimize for the adversarial example (Sec. 3.1 and 3.2). Exploring the adversarial space of such multiple discriminative pathways in a self-ensemble generates highly transferable adversarial examples, as we show next.

### 3.1 SELF-ENSEMBLE: DISCRIMINATIVE PATHWAYS OF VISION TRANSFORMER

A ViT model (Dosovitskiy et al., 2020; Touvron et al., 2020), $\mathcal{F}$, with $n$ transformer blocks can be defined as $\mathcal{F} = (f_1 \circ f_2 \circ f_3 \circ \ldots f_n) \circ g$, where $f_i$ represents a single ViT block comprising of multi-head self-attention and feed-forward layers and $g$ is the final classification head. To avoid notation clutter, we assume that $g$ consists of the final local norm and MLP-head (Touvron et al., 2020; Dosovitskiy et al., 2020). Self-attention layer within the vision transformer model takes a sequence of $m$ image patches as input and outputs the processed patches. We will refer to the representations associated with the sequence of image patches as patch tokens, $P_t \in \mathbb{R}^{m \times d}$ (where $d$ is the dimensionality of each patch representation). Attention in ViT layers is driven by minimizing the empirical risk during training. In the case of classification, patch tokens are further appended with the class token ($Q_t \in \mathbb{R}^{1 \times d}$). These patch and class tokens are refined across multiple blocks ($f_i$) and attention in these layers is guided such that the most discriminative information from patch tokens is preserved within the class token. The final class token is then projected to the number of classes by the classifier, $g$. Due to the availability of class token at each transformer block, we can create an

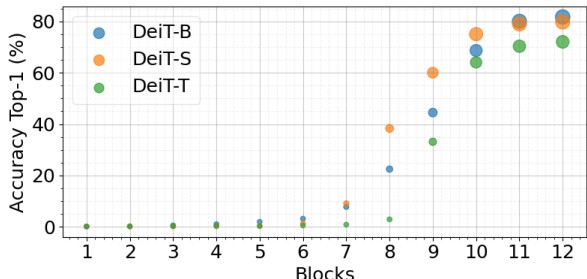

Figure 3: Distribution of discriminative information across blocks of DeiT models. Note how multiple intermediate blocks contain features with considerable discriminative information as measured by top-1 accuracy on the ImageNet val. set. These are standard models pretrained on ImageNet with no further training. Each block (x-axis) corresponds to a classifier $\mathcal{F}_k$ as defined in Equation 1.

ensemble of classifiers by learning a shared classification head at each block along the ViT hierarchy. This provides us an ensemble of $n$ classifiers from a single ViT, termed as the *self-ensemble*:

$$\mathcal{F}_k = \left(\prod_{i=1}^{k} f_i\right) \circ g, \quad \text{where } k = 1, 2, \ldots, n. \tag{1}$$

We note that the multiple classifiers thus formed hold significant discriminative information. This is validated by studying the classification performance of each classifier (Eq. 1) in terms of top-1 (%) accuracy on ImageNet validation set, as demonstrated in Fig. 3. Note that multiple intermediate layers perform well on the task, especially towards the end of the ViT processing hierarchy.

For an input image $x$ with label $y$, an adversarial attack can now be optimized for the ViT's self-ensemble by maximizing the loss at each ViT block. However, we observe that initial blocks (1-6) for all considered DeiT models do not contain useful discriminative information as their classification accuracy is almost zero (Fig. 3). During the training of ViT models (Touvron et al., 2020; Yuan et al., 2021; Mao et al., 2021), parameters are updated based on the last class token only, which means that the intermediate tokens are not directly aligned with the final classification head, $g$ in our self-ensemble approach (Fig. 3) leading to a moderate classification performance. To resolve this, we introduce a token refinement strategy to align the class tokens with the final classifier, $g$, and boost their discriminative ability, which in turn helps improve attack transferability.

## 3.2 TOKEN REFINEMENT

As mentioned above, the multiple discriminative pathways within a ViT give rise to an ensemble of classifiers (Eq. 1). However, the class token produced by each attention layer is being processed by the final classifier, $g$. This puts an upper bound on classification accuracy for each token which is lower than or equal to the accuracy of the final class token. Our objective is to push the accuracy of the class tokens in intermediate blocks towards the upper bound as defined by the last token. For this purpose, we introduced a token refinement module to fine-tune the class tokens.

Our proposed token refinement module is illustrated in Fig. 4. It acts as an intermediate layer inserted between the outputs of each block (after the shared norm layer) and the shared classifier head. Revisiting our baseline ensemble method (Fig. 1), we note that the shared classifier head contains weights directly trained only on the outputs of the last transformer block. While the class tokens of previous layers may be indirectly optimized to align with the final classifier, there exists a potential for misalignment of these features with the classifier: the pretrained classifier (containing weights compatible with the last layer class token) may not extract all the useful information from the previous layers. Our proposed module aims to solve this misalignment by refining the class tokens in a way such that the shared (pretrained) classifier head is able to extract all discriminative information

Figure 4: Recent ViTs process 196 image patches, leading to 196 patch tokens. We rearranged these to create a 14x14 feature grid which is processed by a convolutional block to extract structural information, followed by average pooling to create a single patch token. Class token is refined via a MLP layer before feeding to the classifier. Both tokens are subsequently merged.

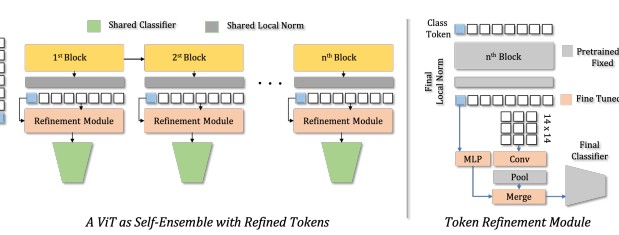

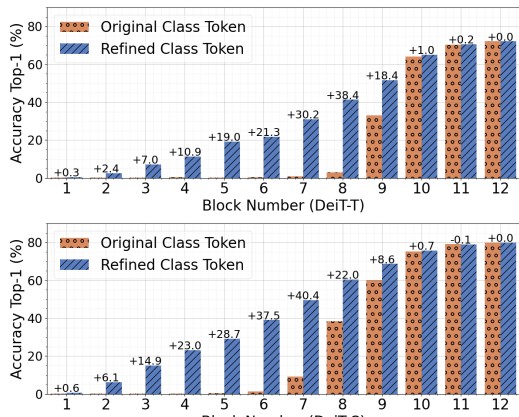

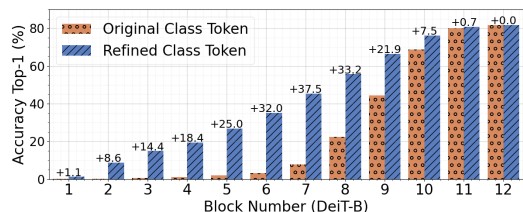

Figure 5: **Self-Ensemble for DeiT (Touvron et al., 2020):** We measure the top-1 accuracy on ImageNet using the class-token of each block and compare to our refined tokens. These results show that fine-tuning helps align tokens from intermediate blocks with the final classifier enhancing their classification performance. Thus token refinement leads to strengthened discriminative pathways allowing more transferable adversaries.

contained within the class tokens of each block. Moreover, intermediate patch tokens may contain additional information that is not at all utilized by the class tokens of those blocks, which would also be addressed by our proposed block. Therefore, we extract both patch tokens and the class token from each block and process them for refinement, as explained next.

−*Patch Token Refinement:* One of the inputs to the token refinement module is the set of patch tokens output from each block. We first rearrange these patch tokens to regain their spatial relationships. The aim of this component within the refinement module is to extract information relevant to spatial structure contained within the intermediate patch tokens. We believe that significant discriminative information is contained within these patches. The obtained rearranged patch tokens are passed through a convolution block (standard ResNet block containing a skip connection) to obtain a spatially aware feature map, which is then average pooled to obtain a single feature vector (of same dimension as the class token). This feature vector is expected to extract all spatial information from patch tokens.

−*Class Token Refinement:* By refining the class tokens of each block, we aim to remove any misalignment between the existing class tokens and the shared (pretrained) classifier head. Also, given how the class token does not contain a spatial structure, we simply use a linear layer to refine it. We hypothesize that refined class token at each block would be much more aligned with the shared classifier head allowing it to extract all discriminative information contained within those tokens.

−*Merging Patch and Class Token:* We obtain the refined class token and the patch feature vector (refined output of patch tokens) and sum them together to obtain a merged token. While we tested multiple approaches for merging, simply summing them proved sufficient.

−*Training:* Given a ViT model containing $k$ transformer blocks, we plugin $k$ instances of our token refinement module to the output of each block as illustrated in Figure 4. We obtain the pretrained model, freeze all existing weights, and train only the $k$ token refinement modules for only a single epoch on ImageNet training set. We used SGD optimizer with learning rate set to 0.001. Training finishes in less than one day on a single GPU-V100 even for a large ViT model such as DeiT-B.

As expected, the trained token refinement module leads to increased discriminability of the class tokens, which we illustrate in Figure 5. Note how this leads to significant boosting of discriminative power especially in the earlier blocks, solving the misalignment problem. We build on this enhanced discriminability of the ensemble members towards better transferability, as explained next.

## 3.3 ADVERSARIAL TRANSFER

Our modifications to ViT models with respect to multiple discriminative pathways and token refinement are exploited in relation to adversarial transfer. We consider black-box attack perturbations that are generated using a source (surrogate) ViT model. The source model is only pretrained on ImageNet, modified according to our proposed approach and is subsequently fine-tuned to update only the token refinement module for a single epoch. We experiment with multiple white-box attacks, generating the adversarial examples using a joint loss over the outputs of each block. The transferability of adversarial examples is tested on a range of CNN and ViT models. Given input sample $x$ and its label $y$, the adversarial object for our self-ensemble (Eq. 1) for the untargeted attack is defined as,

| Source (↓) | Attack | Convolutional | | | | Transformers | | | | |
|---|---|---|---|---|---|---|---|---|---|---|
| **Fast Gradient Sign Method (FGSM) (Goodfellow et al., 2014)** | | | | | | | | | | |
| | | BiT50 | Res152 | WRN | DN201 | ViT-L | T2T-24 | TnT | ViT-S | T2T-7 |
| VGG19$_{bn}$ | FGSM | 23.34 | 28.56 | 33.92 | 33.22 | 13.18 | 10.78 | 12.96 | 25.08 | 29.90 |
| MNAS | FGSM | 23.16 | 39.82 | 40.10 | 44.34 | 16.60 | 22.56 | 25.82 | 34.10 | 48.96 |
| Deit-T | FGSM | 29.74 | 37.10 | 38.86 | 42.40 | 44.38 | 35.42 | 50.58 | 73.32 | 57.62 |
| | FGSM$^E$ | 30.34 | 39.60 | 41.42 | 45.58 | 48.34 | 35.08 | 51.00 | 80.74 | 62.82 |
| | FGSM$^{RE}$ | 30.18$_{(+0.44)}$ | 39.82$_{(+2.7)}$ | 41.26$_{(+2.4)}$ | 46.06$_{(+3.7)}$ | 46.76$_{(+2.4)}$ | 32.68$_{(-2.7)}$ | 48.00$_{(-2.6)}$ | 80.10$_{(+6.8)}$ | 63.90$_{(+6.3)}$ |
| Deit-S | FGSM | 25.44 | 31.04 | 33.58 | 36.28 | 36.40 | 33.41 | 41.00 | 58.78 | 43.48 |
| | FGSM$^E$ | 30.82 | 38.38 | 41.06 | 46.00 | 47.20 | 39.00 | 51.44 | 78.90 | 56.70 |
| | FGSM$^{RE}$ | 34.84$_{(+9.4)}$ | 43.86$_{(+12.8)}$ | 46.26$_{(+12.7)}$ | 51.88$_{(+15.6)}$ | 47.92$_{(+11.5)}$ | 39.86$_{(+6.5)}$ | 55.7$_{(+14.7)}$ | 82.00$_{(+23.2)}$ | 66.20$_{(+22.7)}$ |
| Deit-B | FGSM | 22.54 | 31.58 | 33.86 | 34.96 | 30.50 | 27.84 | 33.08 | 50.24 | 40.50 |
| | FGSM$^E$ | 31.12 | 41.46 | 43.02 | 47.12 | 42.28 | 35.40 | 46.22 | 73.04 | 57.32 |
| | FGSM$^{RE}$ | 35.12$_{(+12.6)}$ | 45.74$_{(+14.2)}$ | 48.46$_{(+14.6)}$ | 52.64$_{(+17.7)}$ | 41.68$_{(+11.1)}$ | 36.60$_{(+8.8)}$ | 49.60$_{(+16.5)}$ | 74.40$_{(+24.2)}$ | 65.92$_{(+25.4)}$ |
| **Projected Gradient Decent (PGD) (Madry et al., 2018)** | | | | | | | | | | |
| VGG19$_{bn}$ | PGD | 19.80 | 28.56 | 33.92 | 33.22 | 5.94 | 10.78 | 12.96 | 13.08 | 29.90 |
| MNAS | PGD | 19.44 | 36.28 | 36.22 | 40.20 | 8.04 | 18.04 | 21.16 | 19.60 | 41.70 |
| Deit-T | PGD | 14.22 | 23.98 | 24.16 | 26.76 | 35.70 | 21.54 | 44.24 | 86.86 | 53.74 |
| | PGD$^E$ | 14.42 | 24.58 | 25.46 | 28.38 | 39.84 | 21.86 | 45.08 | 88.44 | 53.80 |
| | PGD$^{RE}$ | 22.46$_{(+8.24)}$ | 34.64$_{(+10.7)}$ | 37.62$_{(+13.5)}$ | 40.56$_{(+13.8)}$ | 58.60$_{(+22.9)}$ | 26.58$_{(+5.0)}$ | 55.52$_{(+11.3)}$ | 96.34$_{(+9.5)}$ | 66.68$_{(+12.9)}$ |
| Deit-S | PGD | 18.78 | 24.96 | 26.38 | 30.38 | 37.84 | 33.46 | 60.62 | 84.38 | 47.14 |
| | PGD$^E$ | 18.98 | 27.72 | 29.54 | 32.90 | 44.30 | 35.40 | 64.76 | 89.82 | 52.76 |
| | PGD$^{RE}$ | 28.96$_{(+10.2)}$ | 38.92$_{(+14.0)}$ | 42.84$_{(+16.5)}$ | 46.82$_{(+16.4)}$ | 60.86$_{(+23.0)}$ | 40.30$_{(+6.8)}$ | 76.10$_{(+15.5)}$ | 97.32$_{(+12.9)}$ | 71.54$_{(+24.4)}$ |
| Deit-B | PGD | 18.68 | 25.56 | 27.90 | 30.24 | 34.08 | 31.98 | 52.76 | 69.82 | 39.80 |
| | PGD$^E$ | 23.64 | 32.84 | 35.40 | 38.66 | 43.56 | 37.82 | 64.20 | 82.32 | 51.68 |
| | PGD$^{RE}$ | 37.92$_{(+19.2)}$ | 49.10$_{(+23.5)}$ | 53.38$_{(+25.5)}$ | 56.96$_{(+26.7)}$ | 56.90$_{(+22.8)}$ | 45.70$_{(+13.7)}$ | 79.56$_{(+26.8)}$ | 94.10$_{(+24.3)}$ | 74.78$_{(+35.0)}$ |

Table 1: **Fool rate (%)** on 5k ImageNet val. adversarial samples at $\epsilon \leq 16$. Perturbations generated from our proposed self-ensemble with refined tokens from a vision transformer have significantly higher success rate.

$$\max_{\boldsymbol{x'}} \sum_{i=1}^{k} [\![\mathcal{F}_k(\boldsymbol{x'})_{argmax} \neq y]\!], \quad \text{s.t.} \quad \|\boldsymbol{x} - \boldsymbol{x'}\|_p \leq \epsilon, \quad k \in \{1, 2, \ldots, n\} \tag{2}$$

where $[\![\cdot]\!]$ is an indicator function. In the case of target attack, the attacker optimizes the above objective towards a specific target class instead of an arbitrary misclassification.

## 4 EXPERIMENTS

We conduct thorough experimentation on a range of standard attack methods to establish the performance boosts obtained through our proposed transferability approach. We create $\ell_\infty$ adversarial attacks with $\epsilon \leq 16$ and observe their transferability by using the following protocols:

**Source (white-box) models:** We mainly study three vision transformers from DeiT (Touvron et al., 2020) family due to their data efficiency. Specifically, the source models are Deit-T, DeiT-S, and DeiT-B (with 5, 22, and 86 million parameters, respectively). They are trained without CNN distillation. Adversarial examples are created on these models using an existing white-box attack (e.g., FGSM (Goodfellow et al., 2014), PGD (Madry et al., 2018) and MIM (Dong et al., 2018)) and then transferred to the black-box target models.

**Target (black-box) models:** We test the black-box transferability across several vision tasks including classification, detection and segmentation. We consider convolutional networks including BiT-ResNet50 (BiT50) (Beyer et al., 2021), ResNet152 (Res152) (He et al., 2016), Wide-ResNet-50-2 (WRN) (Zagoruyko & Komodakis, 2016), DenseNet201 (DN201) (Huang et al., 2017) and other ViT models including Token-to-Token transformer (T2T) (Yuan et al., 2021), Transformer in Transformer (TnT) (Mao et al., 2021), DINO (Caron et al., 2021), and Detection Transformer (DETR) (Carion et al., 2020) as the black-box target models.

**Datasets:** We use ImageNet training set to fine tune our proposed token refinement modules. For evaluating robustness, we selected 5k samples from ImageNet validation set such that 5 random samples from each class that are correctly classified by ResNet50 and ViT small (ViT-S) (Dosovitskiy et al., 2020) are present. In addition, we conduct experiments on COCO (Lin et al., 2014) (5k images) and PASCAL-VOC12 (Everingham et al., 2012) (around 1.2k images) validation set.

| | | Convolutional | | | | Transformers | | | | |
|---|---|---|---|---|---|---|---|---|---|---|
| **Momemtum Iterative Fast Gradient Sign Method (MIM) (Dong et al., 2018)** | | | | | | | | | | |
| Source ($\downarrow$) | Attack | BiT50 | Res152 | WRN | DN201 | ViT-L | T2T-24 | TnT | ViT-S | T2T-7 |
| VGG19$_{bn}$ | MIM | 36.18 | 46.98 | 54.04 | 57.32 | 12.80 | 21.84 | 25.72 | 28.44 | 47.74 |
| MNAS | MIM | 34.78 | 54.34 | 55.40 | 64.06 | 18.88 | 34.54 | 38.70 | 40.58 | 60.02 |
| Deit-T | MIM | 36.22 | 45.56 | 47.86 | 53.26 | 63.84 | 48.44 | 72.52 | 96.44 | 77.66 |
| | MIM$^E$ | 34.92 | 45.58 | 47.98 | 54.50 | 67.16 | 46.38 | 71.02 | 97.74 | 78.02 |
| | MIM$^{RE}$ | 42.04$_{(+5.8)}$ | 54.02$_{(+8.5)}$ | 58.48$_{(+10.6)}$ | 63.00$_{(+9.7)}$ | 79.12$_{(+15.3)}$ | 49.86$_{(+1.4)}$ | 77.80$_{(+5.3)}$ | 99.14$_{(+2.7)}$ | 85.50$_{(+7.8)}$ |
| Deit-S | MIM | 38.32 | 45.06 | 47.90 | 52.66 | 63.38 | 58.86 | 79.56 | 94.22 | 68.00 |
| | MIM$^E$ | 40.66 | 49.52 | 52.98 | 58.40 | 71.78 | 61.06 | 84.42 | 98.12 | 74.58 |
| | MIM$^{RE}$ | 53.70$_{(+15.4)}$ | 61.72$_{(+16.7)}$ | 65.10$_{(+17.2)}$ | 71.74$_{(+19.1)}$ | 84.30$_{(+20.9)}$ | 66.32$_{(+7.5)}$ | 92.02$_{(+12.5)}$ | 99.42$_{(+5.2)}$ | 89.08$_{(+21.1)}$ |
| Deit-B | MIM | 36.98 | 44.66 | 47.98 | 52.14 | 57.48 | 54.40 | 70.84 | 84.74 | 59.34 |
| | MIM$^E$ | 45.30 | 54.30 | 58.34 | 63.32 | 70.42 | 61.84 | 82.80 | 94.46 | 73.66 |
| | MIM$^{RE}$ | 61.58$_{(+24.6)}$ | 70.18$_{(+25.5)}$ | 74.08$_{(+26.1)}$ | 79.12$_{(+27.0)}$ | 81.28$_{(+23.8)}$ | 69.6$_{(+15.2)}$ | 92.20$_{(+21.4)}$ | 94.10$_{(+9.4)}$ | 89.72$_{(+30.4)}$ |
| **MIM with Input Diversity (DIM) (Xie et al., 2019)** | | | | | | | | | | |
| VGG19$_{bn}$ | DIM | 46.90 | 62.08 | 68.30 | 73.48 | 16.86 | 30.16 | 34.70 | 35.42 | 58.62 |
| MNAS | DIM | 43.74 | 62.08 | 68.30 | 73.48 | 25.06 | 42.92 | 47.24 | 52.74 | 71.98 |
| Deit-T | DIM | 57.56 | 68.30 | 70.06 | 77.18 | 62.00 | 70.16 | 82.68 | 89.16 | 86.18 |
| | DIM$^E$ | 60.14 | 70.06 | 69.84 | 78.00 | 66.38 | 72.30 | 85.98 | 93.72 | 90.78 |
| | DIM$^{RE}$ | 62.10$_{(+4.5)}$ | 70.78$_{(+2.5)}$ | 70.78$_{(+0.7)}$ | 78.40$_{(+1.2)}$ | 67.58$_{(+5.6)}$ | 68.56$_{(-1.6)}$ | 84.18$_{(+1.5)}$ | 93.36$_{(+4.2)}$ | 91.52$_{(+5.3)}$ |
| Deit-S | DIM | 59.00 | 62.12 | 63.42 | 67.30 | 62.62 | 73.84 | 79.50 | 82.32 | 74.20 |
| | DIM$^E$ | 68.82 | 74.44 | 75.34 | 80.14 | 76.22 | 84.10 | 91.92 | 94.92 | 88.42 |
| | DIM$^{RE}$ | 76.14$_{(+17.1)}$ | 81.30$_{(+19.18)}$ | 82.64$_{(+19.22)}$ | 86.98$_{(+19.68)}$ | 78.88$_{(+16.3)}$ | 85.26$_{(+11.4)}$ | 93.22$_{(+13.7)}$ | 96.56$_{(+14.2)}$ | 93.60$_{(+19.4)}$ |
| Deit-B | DIM | 56.24 | 59.14 | 60.64 | 64.44 | 61.38 | 69.54 | 73.96 | 76.32 | 64.44 |
| | DIM$^E$ | 73.04 | 78.36 | 80.28 | 83.70 | 79.06 | 85.10 | 91.84 | 94.38 | 86.96 |
| | DIM$^{RE}$ | 80.10$_{(+23.9)}$ | 84.92$_{(+25.8)}$ | 86.36$_{(+25.7)}$ | 89.24$_{(+24.8)}$ | 78.90$_{(+17.5)}$ | 84.00$_{(+14.5)}$ | 92.28$_{(+18.3)}$ | 95.26$_{(+18.9)}$ | 93.42$_{(+28.9)}$ |

Table 2: **Fool rate (%)** on 5k ImageNet val. adversarial samples at $\epsilon \leq 16$. Perturbations generated from our proposed self-ensemble with refined tokens from a vision transformer have significantly higher success rate.

**Evaluation Metrics:** We report fooling rate (percentage of samples for which the predicted label is flipped after adding adversarial perturbations) to evaluate classification. In the case of object detection, we report the decrease in mean average precision (mAP) and for automatic segmentation, we use the popular Jaccard Index. Given the pixel masks for the prediction and the ground-truth, it calculates the ratio between the pixels belonging to intersection and the union of both masks.

**Baseline Attacks:** We show consistent improvements for single step fast gradient sign method (FGSM) (Goodfellow et al., 2014) as well as iterative attacks including PGD (Madry et al., 2018), MIM (Dong et al., 2018) and input diversity (transformation to the inputs) (DIM) (Xie et al., 2019) attacks. Iterative attacks ran for 10 iterations and we set transformation probability for DIM to default 0.7 (Xie et al., 2019). Our approach is not limited to specific attack settings, but existing attacks can simply be adopted to our self-ensemble ViTs with refined tokens. Refer to appendices A-J for extensive analysis with more ViT designs, attacks, datasets (CIFAR10 & Flowers), computational cost comparison, and latent space visualization of our refined token.

## 4.1 CLASSIFICATION

In this section, we discuss the experimental results on adversarial transferability across black-box classification models. For a given attack method '*Attack*', we refer '*Attack*$^E$' and '*Attack*$^{RE}$' as self-ensemble and self-ensemble with refined tokens, respectively, which are the two variants of our approach. We observe that adversarial transferability from ViT models to CNNs is only moderate for conventional attacks (Tables 1 & 2). For example, perturbations found via iterative attacks from DeiT-B to Res152 has even lower transfer than VGG19$_{bn}$. However, the same attacks when applied using our proposed ensemble strategy (Eq. 1) with refined tokens consistently showed improved transferability to other convolutional as well as transformer based models. We observe that models without inductive biases that share architecture similarities show higher transfer rate of adversarial perturbations among them (e.g., from DeiT to ViT (Dosovitskiy et al., 2020)). We further observe that models trained with the same mechanism but lower parameters are more vulnerable to black-box attacks. For example, ViT-S and T2T-T are more vulnerable than their larger counterparts, ViT-L and T2T-24. Also models trained with better strategies that lead to higher generalizability are less vulnerable to black-box attacks e.g., BiT50 is more robust than ResNet152 (Tables 1 and 2).

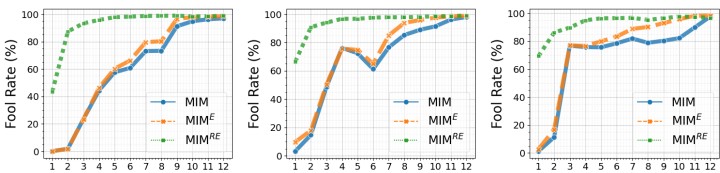

Figure 6: **Ablative Study**: Fooling rate of intermediate layers under MIM (white-box) attack using our self-ensemble approach. We obtain favorable improvements for our method.

| Source (→) | DeiT-T | | DeiT-S | | DeiT-B | |
|---|---|---|---|---|---|---|
| No Attack | MIM | MIM$^{RE}$ | MIM | MIM$^{RE}$ | MIM | MIM$^{RE}$ |
| | 24.0 | 19.7 | 23.7 | 19.0 | 22.9 | 16.9 |
| 38.5 | DIM | DIM$^{RE}$ | DIM | DIM$^{RE}$ | DIM | DIM$^{RE}$ |
| | 20.5 | 13.7 | 20.3 | 12.0 | 19.9 | 11.1 |

Table 3: **Cross-Task Transferability** (*classification→detection*) Object Detector DETR (Carion et al., 2020) is fooled. mAP at [0.5:0.95] IOU on COCO val. set. Our self-ensemble approach with refined token (RE) significantly improves cross-task transferability. (*lower the better*)

| Source (→) | DeiT-T | | DeiT-S | | DeiT-B | |
|---|---|---|---|---|---|---|
| No Attack | MIM | MIM$^{RE}$ | MIM | MIM$^{RE}$ | MIM | MIM$^{RE}$ |
| | 32.5 | 31.6 | 32.5 | 31.0 | 32.6 | 30.6 |
| 42.7 | DIM | DIM$^{RE}$ | DIM | DIM$^{RE}$ | DIM | DIM$^{RE}$ |
| | 31.9 | 31.4 | 31.7 | 31.3 | 32.0 | 31.0 |

Table 4: **Cross-Task Transferability** (*classification→segmentation*) DINO (Caron et al., 2021) is fooled. Jaccard index metric is used to evaluate segmentation performance. Best adversarial transfer results are achieved using our method. (*lower the better*)

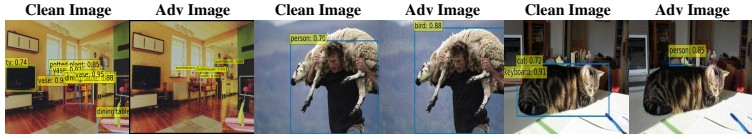

Figure 7: Visualization of DETR failure cases for our proposed DIM$^{RE}$ attack generated from DeiT-S source model. (*best viewed in zoom*)

The strength of our method is also evident by blockwise fooling rate in white-box setting (Fig. 6). It is noteworthy how MIM fails to fool the initial blocks of ViT, while our approach allows the attack to be as effective in the intermediate blocks as for the last class token. This ultimately allows us to fully exploit ViT's adversarial space leading to high transfer rates for adversarial perturbations.

## 4.2 CROSS-TASK TRANSFERABILITY

Self-attention is the core component of transformer architecture regardless of the task; classification (Dosovitskiy et al., 2020; Touvron et al., 2020; Yuan et al., 2021; Mao et al., 2021), object detection (Carion et al., 2020), or unsupervised segmentation (Caron et al., 2021). We explore the effectiveness of our proposed method on two additional tasks: object detection (DETR) (Carion et al., 2020) and segmentation (DINO) (Caron et al., 2021). We select these methods considering the use of transformer modules employing the self-attention mechanism within their architectures. While the task of object detection contains multiple labels per image and involves bounding box regression, the unsupervised model DINO is trained in a self-supervised manner with no traditional image-level labels. Moreover, DINO uses attention maps of a ViT model to generate pixel-level segmentations, which means adversaries must disrupt the entire attention mechanism to degrade its performance.

We generate adversarial signals on source models with a classification objective using their initial predictions as the label. In evaluating attacks on detection and segmentation tasks at their optimal setting, the source ViT need to process images of different sizes (e.g., over 896×896 pix for DETR). To cater for this, we process images in parts (refer appendix G) which allows generation of stronger adversaries. The performance degradation of DETR and DINO on generated adversaries are summarised in Tables 3 & 4. For DETR, we obtain clear improvements. In the more robust DINO model, our transferability increases well with the source model capacity as compared to the baseline.

## 5 CONCLUSION

We identify a key shortcoming in the current state-of-the-art approaches for adversarial transferability of vision transformers (ViTs) and show the potential for much strong attack mechanisms that would exploit the architectural characteristics of ViTs. Our proposed novel approach involving multiple discriminative pathways and token refinement is able to fill in these gaps, achieving significant performance boosts when applied over a range of state-of-the-art attack methods.

**Reproducibility Statement**: Our method simply augments the existing attack approaches and we used open source implementations. We highlight the steps to reproduce all of the results presented in our paper, **a) Attacks:** We used open source implementation of patchwise attack (Gao et al., 2020) and Auto-Attack (Croce & Hein, 2020b) (refer Appendix B) with default setting. Wherever necessary, we clearly mention attack parameters, e.g., iterations for PGD (Madry et al., 2018), MIM (Dong et al., 2018) and DIM (Xie et al., 2019) in section 4 (Experiments: Baseline Attack). Similarly, transformation probability for DIM is set to the default value provided by the corresponding authors that is 0.7, **b) Refined Tokens:** We fine tuned class tokens for the pretrained source models (used to create perturbations) using open source code base (`https://github.com/pytorch/examples/tree/master/imagenet`). We provided training details for fine tuning in section 3.2. Further, we will publicly release all the models with refined tokens, **c) Cross-Task Attack Implementation:** We provided details in section 4.2 and pseudo code in appendix G for cross-task transferability (from classification to segmentation and detection), and **d) Dataset:** We describe the procedure of selecting subset (5k) samples from ImageNet val. set in section 4. We will also release indices of these samples to reproduce the results.

**Ethics Statement**: Since our work focuses on improving adversarial attacks on models, in the short run our work can assist various parties with malicious intents of disrupting real-world deployed deep learning systems dependent on ViTs. However, irrespective of our work, the possibility of such threats emerging exists. We believe that in the long run, works such as ours will support further research on building more robust deep learning models that can withstand the kind of attacks we propose, thus negating the short term risks. Furthermore, a majority of the models used are pre-trained on ImageNet (ILSVRC'12). We also conduct our evaluations using this dataset. The version of ImageNet used contains multiple biases that portray unreasonable social stereotypes. The data contained is mostly limited to the Western world, and encodes multiple gender / ethnicity stereotypes Yang et al. (2020) while also posing privacy risks due to unblurred human faces. In future, we hope to use the more recent version of ImageNet Yang et al. (2021) which could address some of these issues.

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

# Appendix

Adversarial perturbations found with ensemble of models are shown to be more transferable (Dong et al., 2018; Naseer et al., 2021a). In appendix A, we demonstrate the effectiveness of our self-ensemble to boost adversarial transferability within ensemble of different models. Our approach augments and enhances the transferability of the existing attack. We further demonstrate this with recent attacks, Patchwise (Gao et al., 2020) and Auto-Attack (Croce & Hein, 2020b) in appendix B. Auto-Attack is a strong white-box attack which is a combination of PGD with novel losses and other attacks such as boundary-based, FAB (Croce & Hein, 2020a) and, query based Square attack (Andriushchenko et al., 2020). This highlights that our method can be used as plug-and-play with existing attack methods. We then evaluate adversarial transferability of Auto-Attack and PGD (100 iterations) at $\epsilon$ 4, 8, and 16 in appendix C. Our self-ensemble with refined tokens approach consistently performs better with these attack settings as well. In appendix D, we extended our approach to other dataset including CIFAR10 (Krizhevsky et al., 2009) and Flowers (Nilsback & Zisserman, 2008) datasets. We highlight vulnerability of Swin transformer (Liu et al., 2021) against our approach in appendix E. We showed the results on full ImageNet validation set (50k samples) in appendix F. This demonstrates the effectiveness of our method regardless of dataset or task. We discuss the generation of adversarial samples for images of sizes that are different to the source ViT models' input size (e.g., greater than 224) in appendix G. We provide computational cost analysis in Appendix H and visualize the latent space of refined tokens in Appendix H.1. Finally, we extended our approach to diverse ViT designs (Wu et al., 2021; Tolstikhin et al., 2021) and CNN in Appendices I and J.

## A    SELF-ENSEMBLE WITHIN ENSEMBLE

We created an ensemble of pre-trained Deit models (Touvron et al., 2020) including Deit-T, Deit-S and DeiT-B. These models have 12 blocks (layers) and differ in patch embedding size. These models are trained in a similar fashion without distillation from CNN. As expected, adversarial transferability improves with an ensemble of models (Table 5). We also note that unlike such multiple-model ensemble approaches, our self-ensemble attack achieves performance improvements with minimal increase in computational complexity of the attack that is from a single ViT. Our self-ensemble extended three classifiers ensemble to an ensemble of 36 classifiers. Such multi-model ensemble combined with our proposed self-ensemble with refined tokens approach leads to clear attack improvements in terms of transferability (refer Table 5).

| Projected Gradient Decent (PGD) (Madry et al., 2018) | | | | | | | |
|---|---|---|---|---|---|---|---|
| Source (↓) | Attack | Convolutional | | | Transformers | | | |
| | | Res152 | WRN | DN201 | T2T-24 | T2T-7 | TnT | ViT-S |
| Deit-[T+S+B] | PGD | 46.1 | 48.5 | 51.0 | 62.4 | 64.0 | 81.5 | 85.7 |
| | PGD$^E$ | 49.4 | 50.7 | 56.7 | 62.8 | 68.9 | 83.4 | 87.4 |
| | PGD$^{RE}$ | 64.1$_{(+18)}$ | 69.0$_{(+20.5)}$ | 75.1$_{(+24.1)}$ | 73.7$_{(+11.3)}$ | 87.0$_{(+23)}$ | 93.5$_{(+12)}$ | 95.3$_{(+9.6)}$ |
| Momemtum Iterative Fast Gradient Sign Method (MIM) (Dong et al., 2018) | | | | | | | |
| Deit-[T+S+B] | MIM | 62.6 | 66.6 | 70.2 | 77.2 | 76.2 | 86.4 | 87.4 |
| | MIM$^E$ | 67.3 | 68.0 | 73.5 | 76.2 | 78.7 | 87.9 | 87.9 |
| | MIM$^{RE}$ | 76.7$_{(+14.1)}$ | 80.6$_{(+14)}$ | 84.0$_{(+13.8)}$ | 84.0$_{(+6.8)}$ | 89.3$_{(+13.1)}$ | 93.1$_{(+6.7)}$ | 93.6$_{(+6.2)}$ |
| MIM with Input Diversity (Xie et al., 2019) | | | | | | | |
| Deit-[T+S+B] | DIM | 77.2 | 77.9 | 79.7 | 80.7 | 81.4 | 85.0 | 85.7 |
| | DIM$^E$ | 77.8 | 78.0 | 80.7 | 84.6 | 81.5 | 86.1 | 86.4 |
| | DIM$^{RE}$ | 83.1$_{(+5.9)}$ | 84.8$_{(+6.9)}$ | 86.0$_{(+6.3)}$ | 84.9$_{(+4.2)}$ | 87.2$_{(+5.8)}$ | 87.3$_{(+2.3)}$ | 89.0$_{(+3.3)}$ |

Table 5: **Self-ensemble within Ensemble:** Fool rate (%) on 5k ImageNet val. adversarial samples at $\epsilon \leq 16$. Our proposed self-ensemble with refined tokens have significantly higher success rate of adversarial perturbations found within an ensemble of models (Deit-T, Deit-S and Deit-B).

## B  SELF-ENSEMBLE FOR MORE ATTACKS: PATCHWISE AND AUTO-ATTACK

We conducted additional experiments with recent methods such as Patch-wise attack Gao et al. (2020) and Auto-Attack Croce & Hein (2020b) to show case that our approach does not need special setup and can be used as plug-and-play with the existing attacks. Patch-wise Gao et al. (2020) is a recent black-box attack while Auto-Attack is a combination of PGD with novel losses and other attacks such as boundary-based, FAB (Croce & Hein, 2020a) and, query based Square attack (Andriushchenko et al., 2020). In each case, attack transferability is increased significantly using our approach of self-ensemble with refined tokens (refer Table 6).

| Source (↓) | Attack | Convolutional | | | Transformers | | | |
|---|---|---|---|---|---|---|---|---|
| | | Res152 | WRN | DN201 | T2T-24 | T2T-7 | TnT | ViT-S |
| | | **Patchwise (PI) attack (Gao et al., 2020)** | | | | | | |
| Deit-T | PI | 54.1 | 58.5 | 62.6 | 41.1 | 65.3 | 55.6 | 87.0 |
| | $PI^E$ | 56.0 | 61.3 | 65.3 | 42.8 | 68.8 | 58.0 | 89.9 |
| | $PI^{RE}$ | $60.4_{(+6.3)}$ | $67.3_{(+8.8)}$ | $71.4_{(+8.8)}$ | $44.1_{(+3)}$ | $74.8_{(+9.5)}$ | $64.0_{(+8.4)}$ | $95.6_{(+8.6)}$ |
| Deit-S | PI | 53.1 | 57.6 | 63.9 | 49.6 | 59.7 | 64.1 | 84.2 |
| | $PI^E$ | 57.0 | 62.6 | 66.8 | 51.8 | 65.0 | 71.6 | 89.5 |
| | $PI^{RE}$ | $63.6_{(+10.5)}$ | $70.4_{(+12.8)}$ | $74.8_{(+10.9)}$ | $53.1_{(+3.5)}$ | $76.3_{(+16.6)}$ | $78.4_{(+14.3)}$ | $96.2_{(+12)}$ |
| Deit-B | PI | 53.3 | 56.6 | 60.2 | 47.8 | 54.0 | 59.8 | 74.7 |
| | $PI^E$ | 60.7 | 64.4 | 70.0 | 53.0 | 63.3 | 71.4 | 86.6 |
| | $PI^{RE}$ | $70.0_{(+16.6)}$ | $73.8_{(+17.2)}$ | $77.8_{(+17.6)}$ | $57.7_{(+9.9)}$ | $77.3_{(+23.3)}$ | $78.4_{(+18.6)}$ | $94.0_{(+19.3)}$ |
| | | **Auto-Attack (AA) (Croce & Hein, 2020b)** | | | | | | |
| Deit-T | AA | 15.9 | 19.6 | 19.0 | 11.0 | 34.3 | 19.9 | 52.0 |
| | $AA^E$ | 14.0 | 15.8 | 17.1 | 10.3 | 27.9 | 18.0 | 49.3 |
| | $AA^{RE}$ | $20.0_{(+4.1)}$ | $24.2_{(+4.6)}$ | $27.7_{(+8.7)}$ | $13.4_{(+2.4)}$ | $40.0_{(+5.7)}$ | $28.1_{(+8.2)}$ | $58.8_{(+6.8)}$ |
| Deit-S | AA | 23.5 | 22.9 | 24.8 | 21.0 | 40.4 | 36.2 | 61.3 |
| | $AA^E$ | 21.9 | 23.3 | 25.8 | 22.2 | 38.6 | 37.8 | 60.3 |
| | $AA^{RE}$ | $29.3_{(+5.8)}$ | $32.9_{(+10)}$ | $36.7_{(+11.9)}$ | $30.2_{(+9.2)}$ | $51.9_{(+11.5)}$ | $49.7_{(+13.5)}$ | $68.5_{(+7.2)}$ |
| Deit-B | AA | 25.6 | 28.5 | 28.4 | 23.4 | 39.0 | 39.9 | 55.7 |
| | $AA^E$ | 26.6 | 29.2 | 31.5 | 26.1 | 40.7 | 40.2 | 55.8 |
| | $AA^{RE}$ | $37.4_{(+11.8)}$ | $40.4_{(+11.9)}$ | $42.6_{(+14.2)}$ | $36.3_{(+12.9)}$ | $56.7_{(+17.7)}$ | $55.6_{(+15.7)}$ | $69.5_{(+13.8)}$ |

Table 6: **Fool rate** (%) on 5k ImageNet val. adversarial samples at $\epsilon \leq 16$. Auto-attack and patchwise attacks when applied to our proposed self-ensemble with refined tokens have significantly higher transfer rate to unknown convolutional and other vision transformers.

## C  SELF-ENSEMBLE FOR DIFFERENT $\epsilon$: 4, 8, 16

We evaluate if adversarial transferability boost provided by our method also hold for large number of attack iterations such PGD with 100 iterations and Auto-Attack which is based on multiple random restarts, thousands of queries, etc. We further evaluate adversarial transfer at various perturbation budgets, $\epsilon$ such as 4, 8, and 16. Our self-ensemble with refined tokens approach consistently improves black-box adversarial transfer under such attack setting, thus making them not only powerful in white-box but also in black-box setting. Results are presented in Table 7.

## D  SELF-ENSEMBLE FOR MORE DATASETS: CIFAR10 AND FLOWERS

We extended our approach to CIFAR10 and Flowers datasets. Since ViT training from scratch on small datasets is unstable Touvron et al. (2020); Yuan et al. (2021), we fine-tuned ViTs trained on ImageNet on CIFAR10 and Flowers. Transferability of adversarial attacks increases with notable gains by using our approach based on self-ensemble and refined tokens (refer Tables 8, 9, 10 and 11).

| Auto-Attack (Croce & Hein, 2020b) | | | | | | | | |
|---|---|---|---|---|---|---|---|---|
| Source (↓) | Attack | Convolutional | | | Transformers | | | |
| | | Res152 | WRN | DN201 | T2T-24 | T2T-7 | TnT | ViT-S |
| Deit-T | AA | 4.2/9.0/15.9 | 4.3/8.9/19.6 | 5.4/9.3/19.0 | 3.3/4.8/11.0 | 10.6/18.8/34.3 | 5.1/10.3/19.9 | 17.6/33.7/52.0 |
| | $AA^E$ | 4.4/8.9/14.0 | 4.4/8.6/15.8 | 5.0/9.6/17.1 | 3.7/4.8/11.3 | 10.8/17.6/27.9 | 5.7/10.0/18.0 | 18.5/34.4/49.3 |
| | $AA^{RE}$ | **6.3/12.1/20.0** | **7.3/12.9/24.2** | **7.3/13.9/27.7** | **4.7/9.8/13.4** | **15.9/28.3/40.0** | **9.2/15.5/28.1** | **28.7/47.6/58.8** |
| Deit-S | AA | 8.4/12.9/23.5 | 7.9/13.2/22.9 | 8.0/14.7/24.8 | 7.3/11.3/21 | 15.9/25.0/40.4 | 12.4/20.6/36.2 | 20.7/42.0/61.3 |
| | $AA^E$ | 8.8/12.5/21.9 | 7.6/13.6/23.3 | 9.0/15.1/25.8 | 9.1/14.3/22.2 | 16.0/25.2/38.6 | 18.3/27.2/37.8 | 28.9/44.9/60.3 |
| | $AA^{RE}$ | **9.5/18.5/29.3** | **10.5/19.7/32.9** | **12.0/22.9/36.7** | **11.6/18.1/30.2** | **24.6/37.2/51.9** | **23.3/35.2/49.7** | **37.6/58.4/68.5** |
| Deit-B | AA | 8.2/14.4/25.6 | 8.4/13.6/28.5 | 9.8/16.5/28.4 | 7.7/12.4/23.4 | 15.4/22.4/39.0 | 13.2/21.6/39.9 | 16.6/32.8/55.7 |
| | $AA^E$ | 8.8/15.3/26.6 | 9.1/17.4/29.2 | 11.1/19.3/31.5 | 9.5/15.7/26.1 | 17.3/26.3/40.7 | 16.2/27.0/40.2 | 21.3/37.0/55.8 |
| | $AA^{RE}$ | **12.1/21.5/37.4** | **13.3/24.5/40.4** | **14.2/26.7/42.6** | **12.6/21.5/36.3** | **25.0/39.5/56.7** | **26.9/40.6/55.6** | **29.6/53.8/69.5** |
| Projected Gradient Decent (PGD) (Madry et al., 2018) | | | | | | | | |
| Deit-T | PGD | 6.3/15.2/27.5 | 8.1/17.9/30.1 | 8.7/20.3/33.8 | 7.0/9.6/24.2 | 18.6/30.8/56.1 | 13.7/33.6/45.3 | 33.6/63.7/87.8 |
| | $PGD^E$ | 8.5/18.6/30.5 | 9.0/18.2/30.7 | 10.7/19.2/34.5 | 7.9/10.0/26.3 | 23.1/39.8/58.9 | 14.9/35.8/47.1 | 52.3/75.0/90.3 |
| | $PGD^{RE}$ | **12.3/28.9/40.6** | **18.8/23.5/44.0** | **16.5/29.1/48.8** | **15.0/26.8/33.1** | **30.9/53.9/73.1** | **29.2/52.3/69.7** | **68.2/80.1/98.9** |
| Deit-S | PGD | 9.3/16.8/30.1 | 14.9/20.5/32.7 | 9.8/23.5/35.3 | 16.3/20.3/31.5 | 26.9/36.3/51.8 | 26.8/40.6/56.1 | 40.9/62.5/84.5 |
| | $PGD^E$ | 12.3/17.0/35.2 | 16.7/24.1/36.5 | 11.2/26.3/38.9 | 17.0/22.2/34.8 | 32.5/41.2/53.0 | 28.0/44.8/62.1 | 51.2/77.7/90.7 |
| | $PGD^{RE}$ | **17.8/30.1/44.7** | **18.7/33.6/50.6** | **20.7/35.6/54.9** | **23.1/34.1/45.8** | **48.5/67.8/80.6** | **40.1/65.9/85.0** | **67.8/88.8/98.4** |
| Deit-B | PGD | 13.0/20.4/33.2 | 11.5/18.6/33.8 | 14.7/28.9/46.1 | 14.7/19.4/30.0 | 15.3/24.8/37.9 | 20.2/35.2/50.7 | 20.6/44.8/65.2 |
| | $PGD^E$ | 13.5/21.5/40.1 | 17.1/28.4/44.3 | 17.8/30.7/47.8 | 16.7/35.7/45.0 | 26.7/42.2/57.6 | 36.8/60.1/73.3 | 44.8/67.5/85.6 |
| | $PGD^{RE}$ | **19.1/38.9/60.8** | **23.3/40.8/64.7** | **30.2/56.8/70.8** | **19.6/40.3/56.1** | **49.6/66.6/80.2** | **45.9/69.9/88.7** | **59.6/82.6/98.6** |

Table 7: **Fool rate** (%) on 5k ImageNet val. at various perturbation budgets. Auto-attack and PGD (100 iterations) are evaluated at $\epsilon \leq 4/8/16$. Our method consistently performs better.

| Projected Gradient Descent (PGD) (Madry et al., 2018) | | | | |
|---|---|---|---|---|
| Source (↓) | Attack | Transformers | | Convolutional |
| | | T2T-24 | T2T-7 | Res50 |
| Deit-T | PGD | 26.8 / 41.9 | 46.4 / 61.0 | 25.4 / 39.2 |
| | $PGD^E$ | 30.6 / 42.3 | 49.6 / 60.9 | 35.2 / 42.2 |
| | $PGD^{RE}$ | **37.7 / 57.3** | **65.4 / 80.7** | **49.5 / 57.2** |
| Deit-S | PGD | 35.6 / 47.6 | 42.9 / 54.9 | 27.2 / 36.1 |
| | $PGD^E$ | 35.4 / 47.8 | 46.9 / 59.6 | 31.2 / 40.6 |
| | $PGD^{RE}$ | **46.6 / 65.5** | **68.5 / 82.1** | **51.2 / 61.9** |
| Deit-B | PGD | 29.1 / 40.4 | 33.5 / 43.7 | 26.2 / 34.6 |
| | $PGD^E$ | 30.1 / 43.0 | 40.7 / 54.8 | 33.4 / 41.0 |
| | $PGD^{RE}$ | **40.3 / 44.2** | **61.4 / 78.0** | **52.3 / 65.7** |

Table 8: **Self-ensemble with refined tokens for CIFAR10:** PGD fool rate (%) on CIFAR10 test set (10k samples). In each table cell, performances are shown for two perturbation budgets i.e., $\epsilon \leq 8/16$.

# E   VULNERABILITY OF SWIN TRANSFORMER

We show how the black-box vulnerability of ViT architecture without explicit class token (Swin transformer (Liu et al., 2021)) increases against our approach (refer Table 12).

# F   FULL IMAGENET VALIDATION SET

In line with most existing attack methods, we use subset of ImageNet (5k samples, 5 samples from each class) for all our experiments. Our subset from ImageNet is balanced as we sample from each class. We will release the image-indices of ours subset publicly to reproduce the results.

Additionally, we re-run our best performing attacks (MIM and DIM) on the whole ImageNet val. set (50k samples) to validate the merits of our approach ( refer Table 13). Our approach shows considerable improvements in attack transferability regardless the dataset size.

| Momemtum Iterative Fast Gradient Sign Method (MIM) (Dong et al., 2018) | | | | |
|---|---|---|---|---|
| Source (↓) | Attack | Transformers | | Convolutional |
| | | T2T-24 | T2T-7 | Res50 |
| Deit-T | MIM | 45.8 / 70.2 | 63.9 / 81.2 | 50.6 / 72.2 |
| | $MIM^E$ | 48.8 / 70.1 | 65.7 / 81.0 | 57.3 / 74.0 |
| | $MIM^{RE}$ | **57.8 / 83.0** | **78.8 / 90.7** | **57.8 / 77.3** |
| Deit-S | MIM | 62.3 / 79.1 | 62.1 / 79.0 | 48.4 / 69.6 |
| | $MIM^E$ | 62.1 / 78.0 | 65.2 / 82.3 | 52.3 / 71.4 |
| | MIM | **70.3 / 88.8** | **82.0 / 91.4** | **60.2 / 79.3** |
| Deit-B | MIM | 54.3 / 71.1 | 51.6 / 72.7 | 50.6 / 68.7 |
| | $MIM^E$ | 55.4 / 73.8 | 60.8 / 79.6 | 55.6 / 72.4 |
| | $MIM^{RE}$ | **62.3 / 74.8** | **75.8 / 89.7** | **63.6 / 81.3** |
| MIM with Input Diversity (DIM) (Xie et al., 2019) | | | | |
| Deit-T | DIM | 56.4 / 90.1 | 67.1 / 89.1 | 67.4 / 85.4 |
| | $DIM^E$ | 62.6 / 90.2 | 77.8 / 90.3 | 68.4 / 85.6 |
| | $DIM^{RE}$ | **70.2 / 92.2** | **80.4 / 93.6** | **72.5 / 89.0** |
| Deit-S | DIM | 66.9 / 90.5 | 70.8 / 86.5 | 67.7 / 83.8 |
| | $DIM^E$ | 73.7 / 90.8 | 71.2 / 89.2 | 69.7 / 87.2 |
| | $DIM^{RE}$ | **75.9 / 94.5** | **82.6 / 93.7** | **75.3 / 92.6** |
| Deit-B | DIM | 71.7 / 86.9 | 63.6 / 82.4 | 60.3 / 83.3 |
| | $DIM^E$ | 73.0 / 93.6 | 72.4 / 90.3 | 68.7 / 89.8 |
| | $DIM^{RE}$ | **75.7 / 95.0** | **77.5 / 93.0** | **72.4 / 92.8** |

Table 9: **Self-ensemble with refined tokens for CIFAR10:** MIM and DIM fool rate (%) on CIFAR10 test set (10k samples). In each table cell, performances are shown for two perturbation budgets i.e., $\epsilon \leq 8/16$.

| Projected Gradient Descent (PGD) (Madry et al., 2018) | | | | |
|---|---|---|---|---|
| Source (↓) | Attack | Transformers | | Convolutional |
| | | T2T-24 | T2T-7 | Res50 |
| Deit-T | PGD | 22.4 / 28.8 | 31.3 / 41.3 | 16.3 / 26.8 |
| | $PGD^E$ | 25.6 / 35.5 | 35.6 / 48.7 | 25.4 / 33.8 |
| | $PGD^{RE}$ | **26.0 / 37.6** | **39.5 / 55.9** | **38.4 / 47.2** |
| Deit-S | PGD | 20.7 / 26.9 | 30.8 / 41.3 | 17.4 / 26.2 |
| | $PGD^E$ | 23.2 / 31.8 | 34.4 / 47.3 | 25.0 / 32.6 |
| | $PGD^{RE}$ | **25.3 / 35.4** | **40.1 / 56.1** | **43.2 / 52.0** |
| Deit-B | PGD | 20.6 / 26.9 | 32.0 / 41.3 | 18.4 / 27.2 |
| | $PGD^E$ | 23.5 / 32.0 | 34.3 / 48.0 | 29.3 / 36.3 |
| | $PGD^{RE}$ | **27.0 / 39.0** | **39.4 / 56.9** | **49.8 / 57.0** |

Table 10: **Self-ensemble with refined tokens for Flowers:** PGD fool rate (%) on Flowers test set. In each table cell, performances are shown for two perturbation budgets i.e., $\epsilon \leq 8/16$.

# G  CROSS-TASK TRANSFERABILITY WITH LARGER IMAGES (>224)

We generate adversarial signals on source models with a classification objective using their initial predictions as the label (Algorithm 1). In the case of ViT models trained on 224x224 images, we rescale all larger images to a multiple of 224x224, split them into smaller 224x224 image portions, obtain the prediction of the source (surrogate) model for each smaller image portion, and apply the attack separately for each image portion using the prediction as the label for the adversarial objective function (Algorithm 2). For example, in the case of DETR, we use an image size of 896 x 896, split

| Momemtum Iterative Fast Gradient Sign Method (MIM) (Dong et al., 2018) | | | | |
|---|---|---|---|---|
| Source (↓) | Attack | Transformers | | Convolutional |
| | | T2T-24 | T2T-7 | Res50 |
| Deit-T | MIM | 26.3 / 45.2 | 38.1 / 64.6 | 35.8 / 54.6 |
| | MIM$^E$ | 27.5 / 47.9 | 39.4 / 65.9 | 37.8 / 56.9 |
| | MIM$^{RE}$ | **28.0 / 48.6** | **42.2 / 69.5** | **47.2 / 64.9** |
| Deit-S | MIM | 24.1 / 43.5 | 37.0 / 63.3 | 35.3 / 53.2 |
| | MIM$^E$ | 25.4 / 45.6 | 38.3 / 65.5 | 38.6 / 53.8 |
| | MIM$^{RE}$ | **26.7 / 47.1** | **43.0 / 69.6** | **51.8 / 69.5** |
| Deit-B | MIM | 23.7 / 43.4 | 37.8 / 63.4 | 39.2 / 56.3 |
| | MIM$^E$ | 25.6 / 45.2 | 39.0 / 65.0 | 40.3 / 60.0 |
| | MIM$^{RE}$ | **28.7 / 49.2** | **43.7 / 69.3** | **56.3 / 71.2** |
| MIM with Input Diversity (DIM) (Xie et al., 2019) | | | | |
| Deit-T | DIM | 24.2 / 44.4 | 36.5 / 62.6 | 45.8 / 62.5 |
| | DIM$^E$ | 26.0 / 44.7 | 37.9 / 62.9 | 47.2 / 62.8 |
| | DIM$^{RE}$ | **27.3 / 46.9** | **40.1 / 67.1** | **54.7 / 70.0** |
| Deit-S | DIM | 23.1 / 42.0 | 35.7 / 61.5 | 47.2 / 64.1 |
| | DIM$^E$ | 24.0 / 42.8 | 36.1 / 62.2 | 48.1 / 65.3 |
| | DIM$^{RE}$ | **27.1 / 47.0** | **41.5 / 68.7** | **56.3 / 78.6** |
| Deit-B | DIM | 23.4 / 42.9 | 36.3 / 61.5 | 46.3 / 64.3 |
| | DIM$^E$ | 24.4 / 44.5 | 37.1 / 63.4 | 46.4 / 66.0 |
| | DIM$^{RE}$ | **30.0 / 50.2** | **44.3 / 69.4** | **58.4 / 78.8** |

Table 11: **Self-ensemble with refined tokens for Flowers:** MIM and DIM fool rate (%) on Flowers test set. In each table cell, performances are shown for two perturbation budgets i.e., $\epsilon \leq 8/16$.

| Source (↓) | PGD | PGD$^E$ | PGD$^{RE}$ | MIM | MIM$^E$ | MIM$^{RE}$ | DIM | DIM$^E$ | DIM$^{RE}$ |
|---|---|---|---|---|---|---|---|---|---|
| Deit-T | 17.1 | 18.3 | **23.8** | 36.0 | 37.9 | **40.1** | 52.0 | 54.4 | **56.7** |
| Deit-S | 26.7 | 28.2 | **36.2** | 47.1 | 49.9 | **56.0** | 61.5 | 69.6 | **71.8** |
| Deit-B | 29.3 | 35.7 | **46.8** | 49.4 | 57.7 | **66.7** | 60.3 | 74.9 | **77.5** |

Table 12: **Swin Transformer (patch-4, window-7):** Fool rate (%) on 5k ImageNet val. at perturbation budget, $\epsilon \leq 16$. Our method increases the black-box strength of adversarial attacks against Swin Transformer.

it into 16 portions of size 224x224, and use these to individually generate adversarial signals which we later combine to build a joint 896x896 sized adversary.

---

**Algorithm 1** Cross-Task Attack

```
0   for samples, _ in (dataloader):
1       orig_shape = samples.shape
2
3       # run attack
4       with torch.no_grad():
5           clean_out = src_model(samples)
6           label = clean_out.argmax(dim=-1).detach()
7
8       adv = generate_adversary(src_model, samples, label)
9       # done running attack
10
11      samples = adv
12      # continue model evaluation
```

We run a generic attack on cross-tasks where a single image level label is not available. The source model is used to generate a label (its original prediction) which is used as the target for the white-box attack. The generated adversarial signal is then applied onto the target task.

| Momemtum Iterative Fast Gradient Sign Method (MIM) (Dong et al., 2018) | | | | | | | |
|---|---|---|---|---|---|---|---|
| Source (↓) | Attack | Convolutional | | | Transformers | | | |
| | | Res152 | WRN | DN201 | T2T-24 | T2T-7 | TnT | ViT-S |
| Deit-T | MIM | 48.2 | 52.2 | 55.7 | 53.3 | 75.3 | 73.0 | 93.1 |
| | $MIM^E$ | 49.9 | 53.6 | 58.5 | 52.8 | 76.6 | 73.5 | 96.5 |
| | $MIM^{RE}$ | $57.8_{(+9.8)}$ | $62.0_{(+9.8)}$ | $65.2_{(+9.5)}$ | $55.4_{(+2.1)}$ | $84.7_{(+)}$ | $78.8_{(+9.4)}$ | $99.5_{(+6.4)}$ |
| Deit-S | MIM | 48.3 | 46.3 | 56.4 | 60.5 | 70.3 | 80.2 | 92.5 |
| | $MIM^E$ | 53.1 | 50.7 | 60.2 | 63.3 | 78.0 | 85.6 | 97.1 |
| | $MIM^{RE}$ | $66.8_{(+18.5)}$ | $69.0_{(+22.7)}$ | $75.0_{(+18.6)}$ | $69.1_{(+8.6)}$ | $92.3_{(+22)}$ | $96.1_{(+15.9)}$ | $99.5_{(+7)}$ |
| Deit-B | MIM | 41.6 | 45.0 | 55.9 | 56.5 | 60.0 | 68.4 | 70.2 |
| | $MIM^E$ | 58.3 | 60.3 | 65.8 | 64.9 | 76.3 | 85.6 | 83.0 |
| | $MIM^{RE}$ | $75.9_{(+34.3)}$ | $80.2_{(+35.2)}$ | $84.8_{(+28.9)}$ | $72.3_{(+15.8)}$ | $90.1_{(+30.1)}$ | $97.1_{(+28.7)}$ | $96.5_{(+26.3)}$ |
| MIM with Input Diversity (DIM) (Xie et al., 2019) | | | | | | | |
| Deit-T | DIM | 66.4 | 67.0 | 72.0 | 70.1 | 82.8 | 80.5 | 90.2 |
| | $DIM^E$ | 70.2 | 70.9 | 77.0 | 70.2 | 87.2 | 83.5 | 95.3 |
| | $DIM^{RE}$ | $72.8_{(+6.4)}$ | $74.8_{(+7.8)}$ | $79.7_{(+7.7)}$ | $73.2_{(+3.1)}$ | $89.0_{(+6.2)}$ | $85.4_{(+4.9)}$ | $97.8_{(+7.6)}$ |
| Deit-S | DIM | 60.3 | 60.2 | 65.0 | 74.2 | 67.0 | 81.5 | 80.1 |
| | $DIM^E$ | 77.6 | 76.8 | 81.3 | 88.3 | 89.1 | 92.6 | 95.6 |
| | $DIM^{RE}$ | $82.0_{(+21.7)}$ | $83.1_{(+22.9)}$ | $87.2_{(+22.3)}$ | $89.9_{(+15.7)}$ | $89.1_{(+)}$ | $96.4_{(+22.1)}$ | $98.3_{(+18.2)}$ |
| Deit-B | DIM | 58.4 | 59.2 | 65.3 | 70.5 | 65.3 | 75.0 | 77.2 |
| | $DIM^E$ | 79.6 | 82.7 | 84.8 | 87.3 | 88.3 | 90.4 | 95.7 |
| | $DIM^{RE}$ | $87.0_{(+28.6)}$ | $85.6_{(+26.4)}$ | $91.2_{(+25.9)}$ | $87.0_{(+16.5)}$ | $91.7_{(+26.4)}$ | $93.0_{(+18)}$ | $96.1_{(+18.9)}$ |

Table 13: **Fool rate** (%) on the whole ImageNet val. (50k) adversarial samples at $\epsilon \leq 16$. Our method remains effective and significantly increase adversarial transferability on large dataset as well.

---

**Algorithm 2** Attack for different input sizes

```
from itertools import product

for samples, targets in (dataloader):
    orig_shape = samples.shape

    # run attack
    product_list = product([0, 1, 2, 3], [0, 1, 2, 3])
    temp = torch.cat([
        samples[:,:,224*x:224*(1+x),224*y:224*(1+y)]
            for x, y in product_list], dim=0)

    with torch.no_grad():
        clean_out = src_model(temp)
        label = clean_out.argmax(dim=-1).detach()

    adv = generate_adversary(src_model, temp, label)
    temp = torch.zeros_like(samples)
    for idx, (x, y) in enumerate(product_list):
        temp[:,:,224*x:224*(1+x),224*y:224*(1+y)] =
            adv[orig_shape[0]*idx:orig_shape[0]*(idx+1)]
    adv = temp
    assert orig_shape == adv.shape
    # done running attack
    samples = adv
    # continue model evaluation
```

We run a modified version of the attack for cross-task experiments requiring specific input sizes not compatible with the ViT models' input size. We split larger images into smaller image portions, generate adversarial signals separately for each portion, and combine these to obtain a joint adversarial signal relevant to the entire image. For each image portion, we use the source model to generate a label (its prediction for that image portion), and use it as the target when generating the adversarial signal.

## H    COMPUTATION COST AND INFERENCE SPEED ANALYSIS

In this section, we analyze the parameter complexity and computational cost of our proposed approach. The basic version of our self-ensemble (Attack$^E$) only uses an off the shelf pretrained ViT without adding any additional parameters. However, we introduce additional learnable parameters with token refinement module to solve misalignment problem between the tokens produced by an intermediate

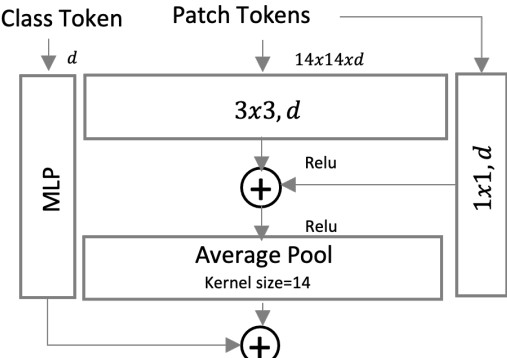

Figure 8: Our proposed refinement module (Sec. 3.2) processes patch tokens using a convolutional block (He et al., 2016) while the class token is processed by a linear layer. Class and patch tokens are the outputs of an intermediate ViT block. The convolutional layers have a filter size of 3x3x$d$. We rearrange the number of patch tokens into 14x14 grid before feeding them to convolutional block. The embedding dimension ($d$) of the class token and each patch token dictates the number of parameters and inference compute cost within convolutional and MLP layers of the refinement module.

| | | | Attacks | | | |
|---|---|---|---|---|---|---|
| **Attack Inference Speed Analysis** | | | | | | |
| Model | Self-ensemble | Refined Tokens | FGSM | PGD | MIM | DIM |
| Deit-T | ✗ | ✗ | 0.19 | 1.48 | 1.49 | 1.51 |
| | ✓ | ✗ | 0.19 | 1.59 | 1.69 | 1.6 |
| | ✓ | ✓ | 0.23 | 2.12 | 2.13 | 2.15 |
| Deit-S | ✗ | ✗ | 0.34 | 3.22 | 3.21 | 3.23 |
| | ✓ | ✗ | 0.36 | 3.34 | 3.32 | 3.35 |
| | ✓ | ✓ | 0.54 | 5.19 | 5.22 | 5.24 |
| Deit-B | ✗ | ✗ | 0.97 | 9.45 | 9.32 | 9.27 |
| | ✓ | ✗ | 1.0 | 9.66 | 9.43 | 9.44 |
| | ✓ | ✓ | 1.64 | 16.15 | 15.88 | 15.9 |

Table 14: We compare inference speed (in minutes) of attacks on the conventional model against the attack on our proposed self-ensemble (with and without refinement module). We used 5k selected samples (Sec. 4) from ImageNet validation set for this experiment and all iterative attacks (PGD, MIM, DIM) ran for 10 iterations. Inference speed is computed using Nvidia Quadro RTX 6000 with Pytorch library.

block and the final classifier of a vision transformer (Sec. 3.2). Our refinement module processes class and patch tokens using MLP (linear layer) and Convolutional block (Fig. 8) and its parameter complexity is dependent on the embedding dimension of these pretrained tokens. For example, DeiT-S produces tokens with embedding dimension of size $\mathbb{R}^{384}$ and our refinement module adds 1.47 million parameters to its sub-model within the self-ensemble trained with refined tokens.

Fine tuning with additional parameters provides notable gains in recognition accuracy. Our approach increases top-1 (%) accuracy (averaged across self-ensemble) by 12.43, 15.21, and 16.70 for Deit-T, Deit-S, and DeiT-B, respectively. Similar trend holds for the convolutional vision transformer (Wu et al., 2021) and MLP-Mixer (Tolstikhin et al., 2021) as well (Fig. 10).

Further, we analyze inference speed for different attacks in Table 14. The computational cost difference between the attack on a conventional model and the attack on our self-ensemble (without refinement module) is only marginal. As expected, attacking self-ensemble with refinement module is slightly more expensive, however, the notable difference is only with very large models such as DeiT-B. In fact, the increase in computational cost is a function of the original complexity of the pre-trained ViT model e.g., DeiT-B with an original parametric complexity of 86 million generates high-dimensional tokens $\mathbb{R}^{784}$, which leads to higher compute cost in our refinement module.

### H.1 LATENT SPACE OF REFINED TOKENS

We visualize the latent space of our refined tokens in Fig. 9. We randomly selected 10 classes from ImageNet validation set distributed across the entire dataset. We extracted class tokens with and without refinement from the intermediate blocks (5,6,7,8) of Deit-T, Deit-S, and Deit-B. Our refined tokens have lower intra-class variations i.e., feature representations of samples from the same class are clustered together. Furthermore, the refined tokens have better inter-class separation than the original tokens. This indicates that refinement minimizes the misalignment between the final classifier and intermediate class tokens, which leads to more disentangled representations. Attacking such disentangled representations across the self-ensemble allows us to find better adversarial direction that leads to more powerful attacks.

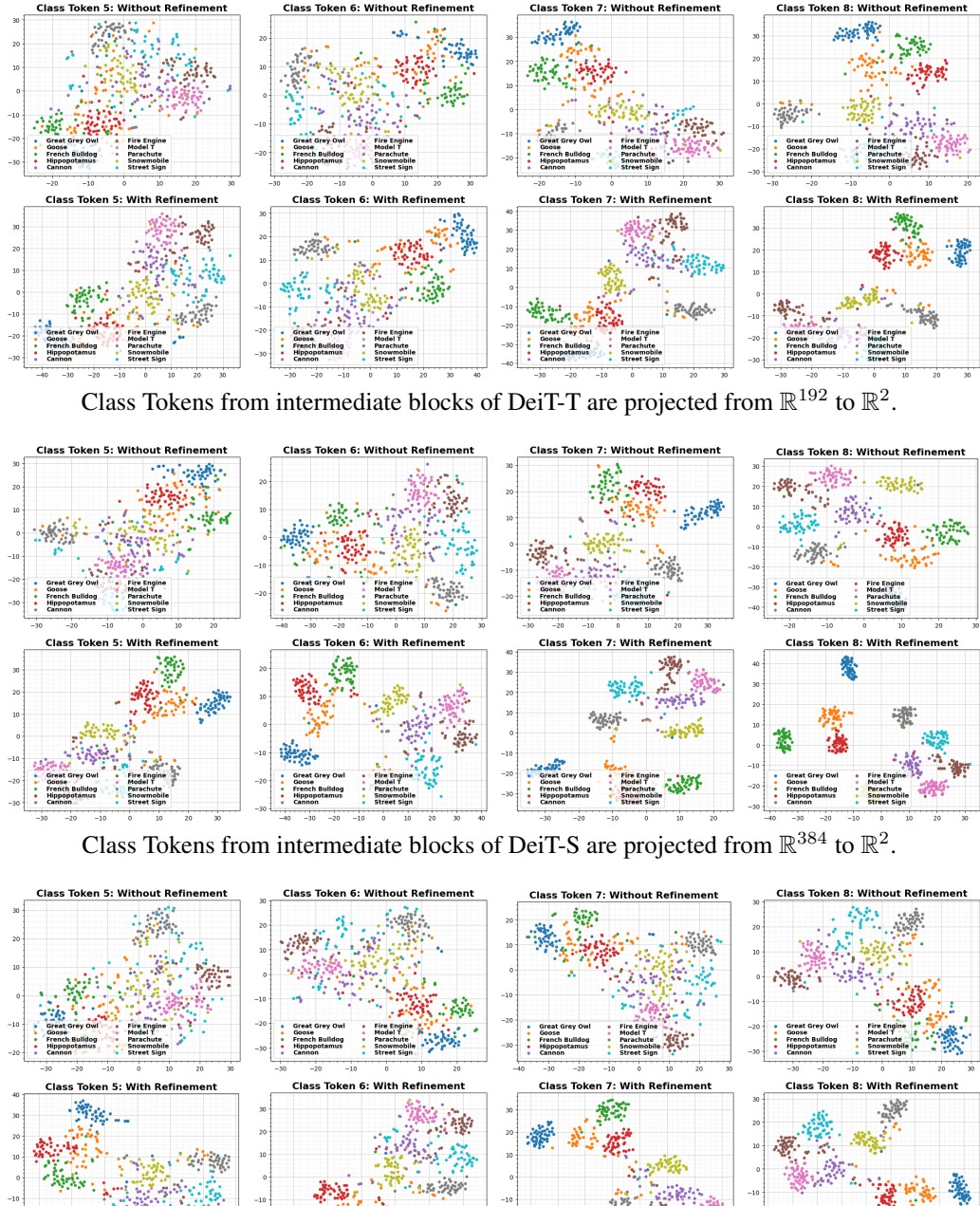

Class Tokens from intermediate blocks of DeiT-T are projected from $\mathbb{R}^{192}$ to $\mathbb{R}^2$.

Class Tokens from intermediate blocks of DeiT-S are projected from $\mathbb{R}^{384}$ to $\mathbb{R}^2$.

Class Tokens from intermediate blocks of DeiT-B are projected from $\mathbb{R}^{784}$ to $\mathbb{R}^2$.

Figure 9: **Class Tokens t-SNE visualization:** We extracted class tokens from the intermediate blocks (used for creating individual models within self-ensemble) and visualize these in 2D space via t-SNE (Van der Maaten & Hinton, 2008). Our refined tokens have lower intraclass variations i.e., feature representations of the same class samples are clustered together. Further refined tokens have better interclass separation than original tokens. We used sklearn (Pedregosa et al., 2011) and perplexity is set to 30 for all the experiments. *(best viewed in zoom)*

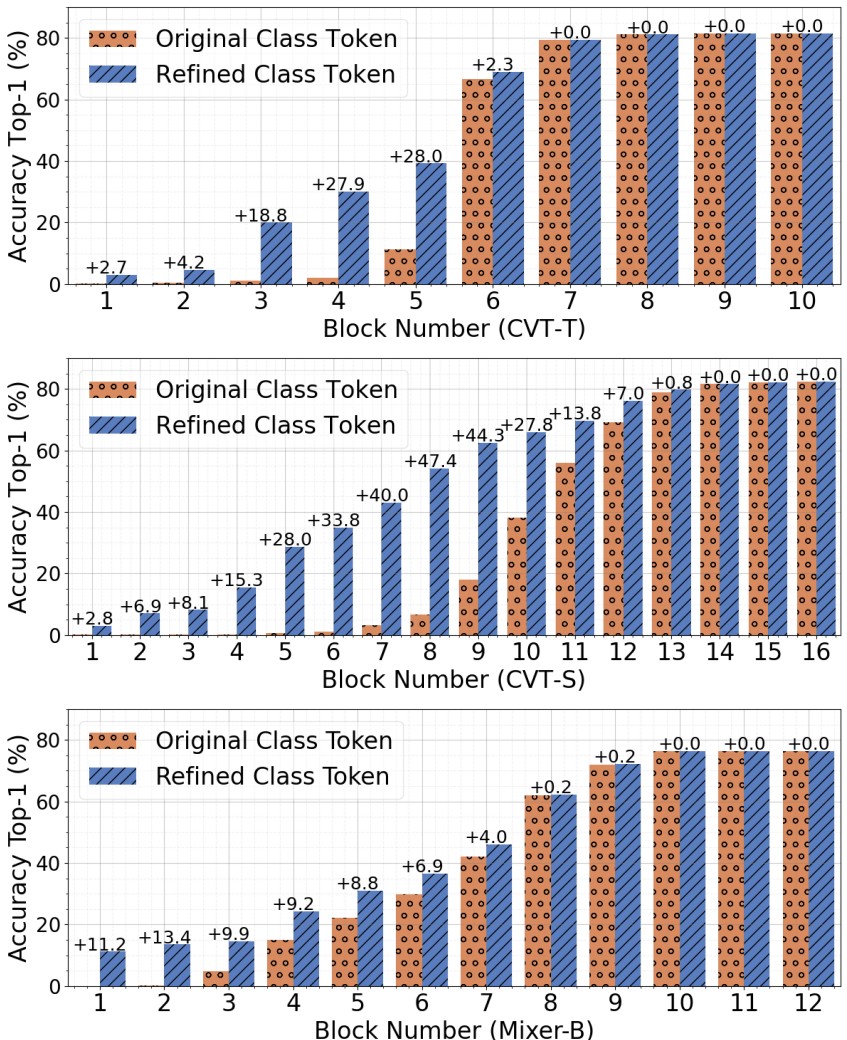

Figure 10: **Self-Ensemble for CvT** (Wu et al., 2021): We measure the top-1 (%) accuracy on ImageNet validation set using the class-token and average of patch tokens of each block for CvT and MLP-Mixer and compare to our refined tokens. These results show that fine-tuning helps align tokens from intermediate blocks with the final classifier enhancing their classification performance. An interesting observation is that pretrained MLP-Mixer has lower final accuracy than CvT models, however, its early blocks show more discriminability than CvT. This allows a powerful self-ensemble which in turn boost adversarial transferability (Table 15).

## I SELF-ENSEMBLE FOR HYBRID VIT AND MLP-MIXER

In this section, we apply our proposed approach to a diverse set of ViT models including a hybrid vision transformer (convolutional vision transformer (CvT) (Wu et al., 2021)) as well as a MLP-Mixer (Tolstikhin et al., 2021). CvT design incorporates convolutional properties into the ViT architecture. We apply our approach on two CvT models, CvT-Tiny (CvT-T) and CvT-Small (CvT-S). We create 10 and 16 models within self-ensemble of CvT-T and CvT-S, respectively. On the other hand, MLP-Mixer thrives on a simplistic architecture design. It does not have self-attention layers or class token. We use an average of patch tokens as class token in this case and create 12 models within self-ensemble of MLP-Mixer-Base (Mixer-B). We train refinement module for each of these models using the same setup as described in Sec. 3.2. The refined class tokens show clear improvements in top-1 (%) accuracy on ImageNet validation set (Fig. 10). Similarly attacking self-ensemble with refined tokens has non-trivial boost in attack transferability (Table 15). The successful application of our approach to such diverse ViT designs highlights the generality of our method.

| Source (↓) | Attack | Convolutional | | | | Transformers | | | | |
|---|---|---|---|---|---|---|---|---|---|---|
| | | **Fast Gradient Sign Method (FGSM) (Goodfellow et al., 2014)** | | | | | | | | |
| | | BiT50 | Res152 | WRN | DN201 | ViT-L | T2T-24 | TnT | ViT-S | T2T-7 |
| CvT-T | FGSM | 23.82 | 28.34 | 29.66 | 30.84 | 16.74 | 26.06 | 28.96 | 30.28 | 41.74 |
| | $FGSM^E$ | 27.66 | 34.42 | 35.58 | 38.66 | 18.26 | 30.98 | 35.38 | 38.42 | 51.46 |
| | $FGSM^{RE}$ | $30.24_{(+6.4)}$ | $37.74_{(+9.4)}$ | $39.44_{(+9.8)}$ | $42.52_{(+11.7)}$ | $18.30_{(+1.6)}$ | $28.52_{(+2.5)}$ | $35.00_{(+6.0)}$ | $39.30_{(+9.0)}$ | $59.50_{(+17.8)}$ |
| CvT-S | FGSM | 19.50 | 24.04 | 26.56 | 27.42 | 14.6 | 21.48 | 22.58 | 25.80 | 36.82 |
| | $FGSM^E$ | 30.0 | 37.30 | 39.08 | 42.04 | 19.48 | 32.38 | 34.56 | 39.24 | 52.92 |
| | $FGSM^{RE}$ | $33.10_{(+13.6)}$ | $40.28_{(+16.2)}$ | $41.84_{(+15.3)}$ | $45.66_{(+18.2)}$ | $19.44_{(+4.8)}$ | $32.12_{(+10.6)}$ | $35.98_{(+13.4)}$ | $41.26_{(+15.5)}$ | $59.38_{(+22.6)}$ |
| Mixer-B | FGSM | 25.00 | 31.02 | 33.00 | 34.54 | 29.28 | 29.24 | 35.08 | 52.44 | 40.86 |
| | $FGSM^E$ | 33.08 | 39.06 | 41.62 | 46.30 | 36.76 | 32.64 | 45.66 | 73.48 | 58.14 |
| | $FGSM^{RE}$ | $35.21_{(+10.2)}$ | $45.26_{(+14.2)}$ | $45.12_{(+12.1)}$ | $48.65_{(+14.11)}$ | $38.54_{(+9.3)}$ | $36.70_{(+7.3)}$ | $47.22_{(+12.1)}$ | $80.44_{(+28.0)}$ | $63.58_{(+22.7)}$ |
| | | **Projected Gradient Decent (PGD) (Madry et al., 2018)** | | | | | | | | |
| CvT-T | PGD | 20.40 | 23.82 | 25.98 | 25.76 | 8.16 | 27.34 | 31.30 | 20.74 | 46.16 |
| | $PGD^E$ | 21.36 | 26.06 | 28.86 | 28.18 | 8.78 | 28.66 | 33.38 | 23.40 | 50.14 |
| | $PGD^{RE}$ | $23.20_{(+2.8)}$ | $28.44_{(+4.6)}$ | $29.56_{(+3.8)}$ | $31.28_{(+5.5)}$ | $8.80_{(+0.6)}$ | $26.92_{(-0.4)}$ | $31.32_{(+0.02)}$ | $22.14_{(+1.4)}$ | $57.70_{(+11.5)}$ |
| CvT-S | PGD | 19.80 | 22.98 | 25.14 | 24.28 | 7.92 | 24.92 | 25.24 | 18.40 | 40.72 |
| | $PGD^E$ | 25.58 | 30.28 | 32.76 | 34.32 | 9.18 | 32.34 | 34.52 | 24.08 | 55.90 |
| | $PGD^{RE}$ | $27.12_{(+7.3)}$ | $30.56_{(+7.6)}$ | $32.28_{(+7.1)}$ | $33.78_{(+9.5)}$ | $8.66_{(+0.7)}$ | $30.22_{(+5.3)}$ | $33.76_{(+8.5)}$ | $23.00_{(+4.6)}$ | $58.68_{(+17.9)}$ |
| Mixer-B | PGD | 12.92 | 16.70 | 17.96 | 19.20 | 14.68 | 16.98 | 23.90 | 42.32 | 29.46 |
| | $PGD^E$ | 24.74 | 32.36 | 35.48 | 37.78 | 28.22 | 31.42 | 49.50 | 76.54 | 58.00 |
| | $PGD^{RE}$ | $26.50_{(+13.6)}$ | $34.28_{(+17.6)}$ | $39.50_{(+21.5)}$ | $38.60_{(+19.4)}$ | $32.56_{(+17.9)}$ | $33.30_{(+16.32)}$ | $54.68_{(+30.8)}$ | $82.90_{(+40.6)}$ | $62.86_{(+33.4)}$ |
| | | **Momemtum Iterative Fast Gradient Sign Method (MIM) (Dong et al., 2018)** | | | | | | | | |
| CvT-T | MIM | 39.30 | 42.68 | 45.94 | 48.88 | 20.38 | 48.74 | 53.50 | 45.46 | 65.94 |
| | $MIM^E$ | 42.24 | 45.66 | 49.88 | 54.02 | 21.46 | 50.28 | 56.26 | 49.46 | 71.60 |
| | $MIM^{RE}$ | $48.92_{(+9.6)}$ | $52.00_{(+9.3)}$ | $55.18_{(+9.24)}$ | $60.22_{(+11.3)}$ | $20.24_{(-0.1)}$ | $50.54_{(+1.8)}$ | $58.74_{(+5.2)}$ | $50.94_{(+5.5)}$ | $81.36_{(+15.4)}$ |
| CvT-S | MIM | 36.60 | 39.94 | 42.64 | 44.8 | 18.48 | 45.06 | 44.92 | 39.12 | 58.64 |
| | $MIM^E$ | 47.04 | 50.72 | 55.02 | 59.56 | 23.06 | 56.06 | 57.84 | 51.20 | 76.54 |
| | $MIM^{RE}$ | $53.26_{(+16.7)}$ | $51.04_{(+11.6)}$ | $55.68_{(+13.0)}$ | $60.22_{(+15.4)}$ | $22.24_{(+3.76)}$ | $55.58_{(+10.5)}$ | $56.80_{(+12.0)}$ | $52.60_{(+13.5)}$ | $79.26_{(+20.6)}$ |
| Mixer-B | MIM | 26.96 | 32.66 | 35.28 | 38.42 | 33.96 | 34.68 | 45.08 | 68.16 | 46.88 |
| | $MIM^E$ | 42.62 | 49.82 | 52.98 | 59.32 | 51.72 | 49.90 | 68.86 | 92.58 | 75.48 |
| | $MIM^{RE}$ | $46.50_{(+19.5)}$ | $55.30_{(+22.6)}$ | $56.20_{(+21.5)}$ | $62.56_{(+24.1)}$ | $53.44_{(+19.5)}$ | $52.00_{(+17.32)}$ | $72.26_{(+27.2)}$ | $94.66_{(+26.5)}$ | $80.12_{(+33.2)}$ |
| | | **MIM with Input Diversity (DIM) (Xie et al., 2019)** | | | | | | | | |
| CvT-T | DIM | 61.94 | 61.60 | 64.22 | 67.72 | 36.94 | 69.80 | 76.20 | 65.70 | 73.68 |
| | $DIM^E$ | 72.22 | 72.94 | 75.88 | 80.80 | 41.54 | 79.24 | 86.12 | 76.02 | 85.62 |
| | $DIM^{RE}$ | $78.02_{(+16.1)}$ | $77.20_{(+15.6)}$ | $80.90_{(+16.7)}$ | $85.74_{(+18.0)}$ | $40.06_{(+3.12)}$ | $78.62_{(+8.82)}$ | $89.34_{(+13.1)}$ | $77.88_{(+12.2)}$ | $92.42_{(+18.7)}$ |
| CvT-S | DIM | 51.64 | 55.28 | 54.16 | 57.22 | 30.98 | 60.1 | 62.36 | 53.30 | 62.46 |
| | $DIM^E$ | 74.96 | 75.88 | 79.56 | 83.5 | 44.06 | 80.94 | 86.40 | 76.86 | 88.06 |
| | $DIM^{RE}$ | $80.45_{(+28.8)}$ | $78.94_{(+23.7)}$ | $81.64_{(+27.5)}$ | $85.74_{(+28.5)}$ | $42.78_{(+11.8)}$ | $82.36_{(+22.3)}$ | $88.50_{(+26.1)}$ | $77.66_{(+24.3)}$ | $92.10_{(+29.6)}$ |
| Mixer-B | DIM | 48.64 | 49.08 | 52.14 | 57.44 | 39.28 | 57.24 | 64.00 | 71.88 | 63.02 |
| | $DIM^E$ | 69.32 | 72.46 | 74.98 | 80.38 | 55.96 | 74.66 | 84.66 | 91.86 | 88.78 |
| | $DIM^{RE}$ | $74.88_{(+26.2)}$ | $76.72_{(+27.6)}$ | $80.10_{(+28.0)}$ | $84.66_{(+27.2)}$ | $60.43_{(+21.2)}$ | $82.17_{(+24.9)}$ | $88.62_{(+24.6)}$ | $95.22_{(+23.3)}$ | $92.92_{(+29.9)}$ |

Table 15: **Fool rate (%)** on 5k ImageNet val. adversarial samples at $\epsilon \leq 16$. Perturbations generated from our proposed self-ensemble with refined tokens from a vision transformer have significantly higher success rate.

## J CAN SELF-ENSEMBLE IMPROVE TRANSFERABILITY FROM CNN?

Our proposed idea of the self-ensemble can be applied to a CNN model (e.g., ResNet50) with an average pooling operation over the intermediate layer outputs. However, due to the varying channel dimension of feature maps across a pretrained CNN , building an ensemble with a shared classifier is not as straight-forward as in ViT (where equidimensional class token is available at each level). For example, ResNet50 has four intermediate blocks that produce feature maps of sizes $\mathbb{R}^{256 \times 56 \times 56}$, $\mathbb{R}^{512 \times 28 \times 28}$, $\mathbb{R}^{1024 \times 14 \times 14}$, and $\mathbb{R}^{2048 \times 7 \times 7}$, respectively. Then, the final classifier processes the average pooled features from the last block. Since there is a mismatch between feature dimensions of the intermediate layers and the final classifier, therefore applying the basic version of our self-ensemble approach (Attack$^E$) is not possible for the pretrained ResNet.

An intermediate layer is essential to project intermediate feature vectors to the same dimension as the final feature vector in the case of CNNs. Therefore, for the refined embedding variant of

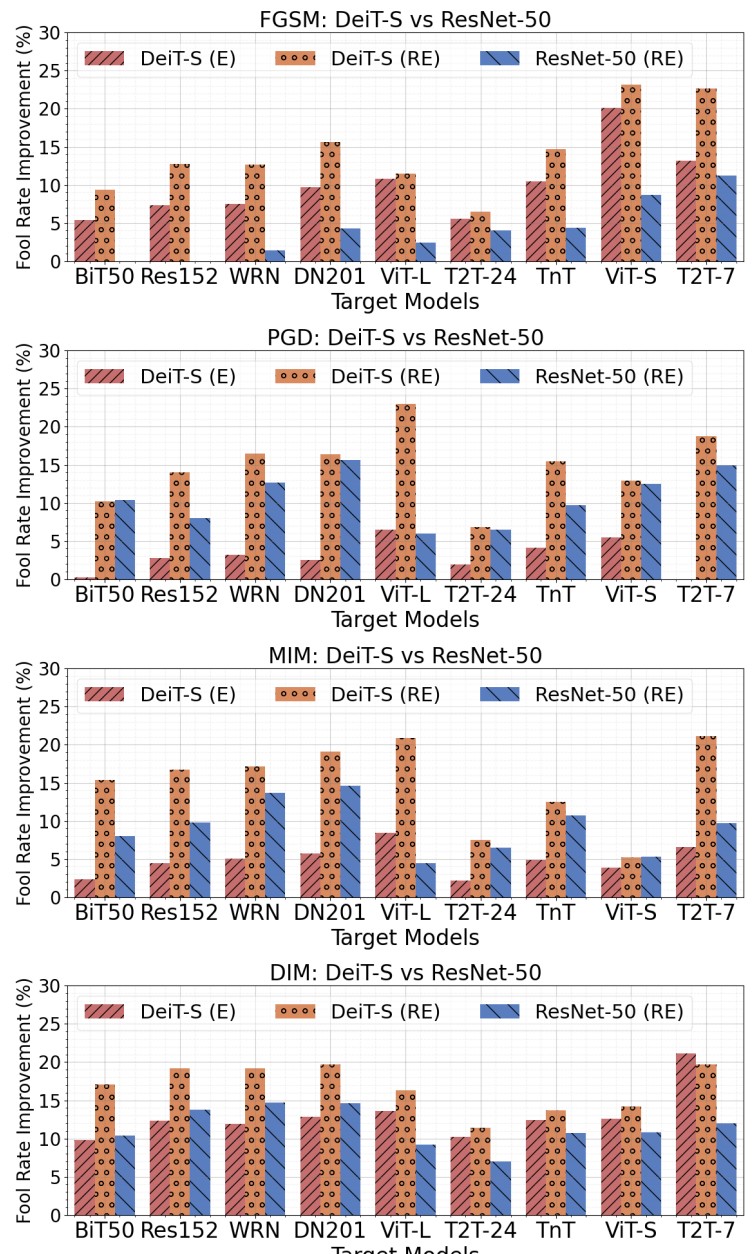

Figure 11: **Self-Ensemble for ResNet50 (**[He et al., 2016](#)**):** We report relative improvement after adding refinement module for Resnet50 and Deit-S mdoels. Note that the basic version of self-ensemble can not applied to ResNet50 due to varying channel dimension across different layers. Feature refinement improves adversarial transfer from ResNet50, however relative gains are significant for vision transformer, Deit-S.

our approach, we finetune ResNet50 features to create self-ensemble with the procedure described in Sec. 3.2. We report the relative improvements of self-ensemble with refined features w.r.t the original (single) model and compare ResNet50 with Deit-S (Fig. 11). Both these off-the-shelf models, ResNet50 (25 million parameters) and Deit-S (22 million parameters), are comparable in terms of their computational complexity. We observe that feature refinement also helps to boost adversarial transferability from the ResNet50 model. However, adversarial transferability improvement of our self-ensemble with token refinement on ViTs is considerably better than for the case of ResNet50.

These results suggest that the proposed approach is well-suited for improving adversarial transferability of ViT models.

