# OpenReview forum: "On Improving Adversarial Transferability of Vision Transformers "
_ICLR.cc/2022/Conference — ICLR 2022 Spotlight_

### Official Review · Reviewer_HKsh · 2021-10-25

**Correctness:** 4
**Technical Novelty And Significance:** 3
**Empirical Novelty And Significance:** 3
**Recommendation:** 8
**Confidence:** 5

**Main Review:**

Strength
1. This is a novel study on ViT adversarial attack modeling, and the self-ensemble method seems to make sense.
2. This work conducts sufficient experiments, and shows good transferablity in a variety of situations.
3. The method of this paper is a general framework. It is not limited to specific calculation method (such as saliency maps and gradients), but is augmented calculation object. So I hope this paper will have a broader impact.

Weekness
1. The author claims that the design of this attack method comes from the different inductive biases between Transformer and CNN style models. However, this paper dosen‘t mention why this inductive bias would cause the previous attack method to fail, nor dose it mention why that difference would make the method more effective.
2. In addition, this method seems to be able to improve the transferablity of models with other paradigms. If this method can improve CNNs' adversarial tranferablity better, it is difficult to say that this is a method of "improving adversarial transferability of ViT model". This paper lacks relevant ablation studys.

**Summary Of The Paper:**

In this paper,  authors improve adversarial transferability of ViT by Self-Ensemble and Token Refinement method.
'Self-Ensemble' implies it treats each transformer block of ViT as the 'last block' and apply a shared classifier to it. So that the trivial discriminative path can be attacked.
'Token Refinement' tries to improve the classification ablity of shallow blocks by using extra non-shared module on each block's output tokens.
This paper successfully improved the adversarial ability of ViT attacks. Sufficient experimental results and detailed description ensure its reproducibility.

**Summary Of The Review:**

In summary, this paper proposes novel methods and shows good reproducibility. Proposed self-ensemble and token reinforcement gets good results in many situations. But I still have doubts about whether this method is for Transformer’s inductive bias.
I am inclined to accept this paper, but I would like to see a more detailed discussion.

---

> ### Author Response · Authors · 2021-11-22
> **Response to HKsh**
>
> We thank the reviewer for the encouraging and insightful comments. Please find our responses to specific queries below.
>
> **On inductive Bias:** Kindly note that the flexible design of ViTs enables learning long-range relationships between pixels, which can be exploited to enhance transferability. Further, the consistent dimensions of ViTs tokens allow leveraging early layers towards creating a self-ensemble, as compared to CNNs that have different dimensional representations across the network hierarchy. Finally, the large receptive fields make even the initial features relatively more discriminative. All these factors contribute towards the proposed approach to be helpful for the ViTs which can mimic the ensemble behavior from a single ViT. Though inductive biases affect transferability rates among different models , however, our approach is equally effective on hybrid ViT designs as well (please see our response to reviewer NMLw: “What effect do surrogate model and black-box target model topological differences have on adversarial transferability?”, and reviewer U2B6: “Hybrid CNN/ViT and Generality of the Proposed Approach”). We will make the discussion more clear in the final manuscript.
>
> **Can the proposed approach improve adversarial transferability from CNN?**
> Our proposed idea of the self-ensemble can be applied to a CNN model (e.g., ResNet50) with an average pooling operation over the intermediate layer outputs. However, due to the varying channel dimension of feature maps across a pretrained CNN [1,2,3], building an ensemble with a shared classifier is not as straightforward as in ViT (where the equidimensional class token is available at each level). For example, ResNet50 has four intermediate blocks that produce feature maps of sizes $\mathbb{R}^{256\times56\times56}$, $\mathbb{R}^{512\times28\times28}$, $\mathbb{R}^{1024\times14\times14}$, and $\mathbb{R}^{2048\times7\times7}$, respectively. Then, the final classifier processes the average pooled features from the last block. Since there is a mismatch between feature dimensions of the intermediate layers and the final classifier, applying the basic version of our self-ensemble approach (Attack$^{E}$) is not possible for the pretrained ResNet.
>
> An intermediate layer is essential to project intermediate feature vectors to the same dimension as the final feature vector in the case of CNNs. Therefore, for the refined embedding variant of our approach, we finetune ResNet50 features to create a self-ensemble with the procedure described in Sec. 3.2. We report the relative improvements of self-ensemble with refined features w.r.t the original (single) model and compare ResNet50 with Deit-S (Fig. 11 in Appendix J). Both these off-the-shelf models, ResNet50 (25 million parameters) and Deit-S (22 million parameters), are comparable in terms of their computational complexity. We observe that feature refinement also helps to boost adversarial transferability from the ResNet50 model. However, adversarial transferability improvement of our self-ensemble with token refinement on ViTs is considerably better than for the case of ResNet50. We have added this discussion in Appendix J.
>
> These results suggest that the proposed approach is well-suited for improving the adversarial transferability of ViT models.
>
> **References**
>
> [1] Simonyan et.al  “Very Deep Convolutional Networks for Large-Scale Image Recognition.” ICLR, 2015
>
> [2] He et.al “Deep Residual Learning for Image Recognition.”, CVPR, 2016
>
> [3] Tan et.al “MnasNet: Platform-Aware Neural Architecture Search for Mobile.”, CVPR 2019

---

> > ### Comment · Reviewer_HKsh · 2021-11-25
> > **Revise recommendation to '8: accept, good paper'**
> >
> > Thank you for your response, which dispel my worries. I think this is a good work, and decide to revise my recommendation.

---

### Official Review · Reviewer_NMLw · 2021-11-04

**Correctness:** 3
**Technical Novelty And Significance:** 3
**Empirical Novelty And Significance:** 3
**Recommendation:** 8
**Confidence:** 3

**Main Review:**

The problem approached by the paper is interesting and the proposed approach is novel to the best of my knowledge. The work is well-structured and written in a clear and concise manner with sufficient experimental verification.

Strengths:
- well written
- sufficient experimental validation including different vision tasks and datasets
- a good overview of previous black-box and white box attacks

Weaknesses:
- will the method work on images with lower dimensions
-  it would have been interesting to include a Future Works section
- a more thorough discussion as why the approach works
- what effect do surrogate model and black-box target model topological differences have on adversarial transferability?
- enhance the quality of images
-

**Summary Of The Paper:**

By incorporating self-ensemble and token refinement methods, the authors have devised a method that takes use of the modularity of ViT models to improve their adversarial transferability. Their experiments have demonstrated adversarial transferability across different convolution and transformer model families for different vision tasks.

**Summary Of The Review:**

Overall, I vote for marginal accept. I like the idea of mining the relation between blocks and handle it by the proposed self-ensemble and token refinement method. My major concern is about the clarity of the paper and discussion as when and why the method works. Hopefully the authors can address my concern in the rebuttal period.

---

> ### Author Response · Authors · 2021-11-22
> **Response to NMLw**
>
> We thank the reviewer for the encouraging and insightful comments. Please find our responses to specific queries below.
>
> **Will the method work on images with lower dimensions?**
> Kindly note that our approach works on images with lower dimensions. We provide detailed experiments on CIFAR10 and Flowers datasets in Appendix D. We will make the reference to the appendices more clear in the final draft.
>
> **Future work:**
> The recent success of ViTs means that the adversarial properties of ViT models also become an important research topic. To the best of our knowledge, we are the first to propose an approach that enhances the block-wise discriminative ability of off-the-shelf vision transformers (ViTs), thereby improving the performance of weak ensembles found within a single vision transformer. Our method allows adversarial perturbations to transfer well from ViTs to other models (ViTs or CNN). An interesting future direction is then to utilize self-ensemble from a single model and  train a ViT that is robust to different blackbox attacks. This can potentially eliminate the need for the ensemble of multiple models required in “Ensemble Adversarial Training”, such as in [1]. Further, refined tokens might also enhance the knowledge transfer from ViTs to downstream tasks [2]. As suggested by the reviewer, we will add this discussion in our final manuscript.
>
> **Why the proposed approach works?**
> The reason why our approach boosts the adversarial transferability is based on ensemble effect with higher discriminative abilities across sub-models. Attacking ensemble of models [3,4] generates more transferable adversaries. This is also highlighted by our own results in Appendix A (Table 5), where attacking ensemble of Deit-T, Deit-S and DeiT-B improves transferability than any of the individual model. Our approach mimics this behaviour with self-ensemble from a single model. Multi-model ensemble when further combined with our proposed self-ensemble with refined tokens approach leads to clear attack improvements in terms of transferability (Appendix A).
>
> **What effect do surrogate model and black-box target model topological differences have on adversarial transferability?**
> Our experimental analysis indicates that  ViTs that are designed on similar principles have higher transferability among them. For example, T2T [5] that is designed to model the local structure in the input images is closer to convolutional vision transformer (CvT) [6], hence the adversarial transferability rates from CvT to T2T models are significantly higher (Table 15 in Appendix I). Similarly, models without inductive biases  that share architecture similarities show higher transfer rate of adversarial perturbations among them (e.g., from DeiT to ViT [6], Tables 1 and 2 in Sec.4). We further observe that  models trained with the same mechanism but lower parameters are more vulnerable to black-box attacks. For example, ViT-S and T2T-7 are more vulnerable than their larger counterparts, ViT-L and T2T-24. Also, models trained with better strategies that lead to higher generalizability are less vulnerable to black-box attacks e.g., BiT50 is more robust than ResNet152. We will add this discussion in the final manuscript.
>
> **Enhance the quality of Images:** Thanks for the suggestion. We will enhance the quality of the images in the final draft.
>
> **References**
>
> [1] Tamer et.al, “Ensemble Adversarial Training: Attacks and Defenses.” ICLR, 2018
>
> [2] “Uncovering the Connections Between Adversarial Transferability and Knowledge Transferability.” ICML, 2021
>
> [3] Dong et al., “Boosting Adversarial Attacks with Momentum.” CVPR, 2018
>
> [4] Xie et al., “Improving Transferability of Adversarial Examples With Input Diversity.” CVPR, 2019
>
> [5] Yuan et.al, “Tokens-to-Token ViT: Training Vision Transformers from Scratch on ImageNet.” ICCV, 2021
>
> [6] Dosovitskiy et.al, “An Image is Worth 16x16 Words: Transformers for Image Recognition at Scale.” ICLR, 2021
>
> [7] Wu et.al, “CvT: Introducing Convolutions to Vision Transformers.” ArXiv, 2021
>
> [8] Tolstikhin et.al, “MLP-Mixer: An all-MLP Architecture for Vision.” ArXiv, 2021

---

### Official Review · Reviewer_U2B6 · 2021-11-04

**Correctness:** 4
**Technical Novelty And Significance:** 3
**Empirical Novelty And Significance:** 4
**Recommendation:** 8
**Confidence:** 3

**Main Review:**

In general, I find the paper presents an interesting and important work w.r.t the adversarial security of ViTs. The main strengths of the paper are:

- The paper is well organized and well-written.
- The proposed approaches are well-motivated by empirical findings.
- The paper has several experiments that the effectiveness of these approaches.

I also have some concerns/questions about the discussions in the paper:

- In my view, the addition of the ensemble module and refined tokens is similar to an approach that combines both CNN and ViT. In Figure 6, it is also evidence that the addition of the refined tokens is important. This begs the question of whether adversarial samples of hybrid CNN/ViT models have similar transferability performance as of the proposed methods in the paper? Currently, it seems like the paper only studies one particular type of ViTs, and the generality of the proposed methods, as well as the generality of the motivation, can be limited.
- Does the same trend follow w.r.t adversarial transferability to other large CNN models such as BiT?



**Summary Of The Paper:**

The paper looks at Vision Transforms (ViTs) models and transferability of adversarial examples, which is previously known to be challenging between ViTs to CNNs and vice versa. The paper leverages the discriminative information stored in the lower layers' tokens and proposes two methods that modify and fine-tune the ViTs in order to extract the adversarial samples. The paper conducts extensive experiments to show the effectiveness in adversarial transferability across different models, including CNN's and ViT's variants, and tasks.

**Summary Of The Review:**

I find that the paper presents an interesting study on the transferability of adversarial samples of ViTs. In general, I think it is a good paper, but also have some concerns about the generality of the work.

---

> ### Author Response · Authors · 2021-11-22
> **Response to U2B6**
>
> We thank the reviewer for the encouraging and insightful comments. Please find our responses to specific queries below.
>
> **Hybrid CNN/ViT and Generality of the Proposed Approach:**
> Our approach is generic in nature and provides benefit to different ViTs regardless of their design. To showcase this,  we apply our approach to hybrid vision transformers such as convolutional vision transformer (CvT) [1] as well as MLP-Mixer [2].  CvT design incorporates convolutional properties into the ViT architecture. We apply our approach on two CvT models, CvT-Tiny and CvT-Small. We created 10 and 16 models within self-ensemble of CvT-Tiny and CvT-Small, respectively. On the other hand, MLP-Mixer thrives on simplistic architecture design. It does not have self-attention layers or class token. We used the average of patch tokens as class token in this case and created 12 models within the self-ensemble of MLP-Mixer-Base. We train the refinement module for each of these sub-models using the same setup as described in Sec. 3.2. The refined class tokens show clear improvements in top-1 (%) accuracy on the ImageNet validation set (Fig. 10 in Appendix I). Similarly attacking self-ensemble with refined tokens has a non-trivial boost in attack transferability (Table 15 in Appendix I).
>
> The successful application of our approach to such diverse ViT designs highlights the generality of the proposed method.
>
> **Does the transferability trend hold for BiT?**
> As recommended, we evaluate BiT [3] model against adversarial attacks. We used the latest best model BiT50 (ResNet50), which is trained by distilling knowledge from BiT-M-R152x2 [4] (trained on 21k classes). BiT50 achieves 82.8% top-1 accuracy on ImageNet validation set. Our analysis (Tables 1 and 2 in Sec. 4.1) shows that BiT50 is more robust than other CNN models but the transferability trends remain the same. As an example, the fooling rate of BiT50 on adversarial examples generated by the DIM attack on DeiT-B increases from 56.24 to 73.04 with self-ensemble and further to 80.10 with refined tokens. Similar trends hold for other attacks (Tables 1 and 2 in Sec. 4.1).
>
> **References**
>
> [1] Wu et.al, “CvT: Introducing Convolutions to Vision Transformers.” ArXiv, 2021
>
> [2] Tolstikhin et.al, “MLP-Mixer: An all-MLP Architecture for Vision.” ArXiv, 2021
>
> [3] Beyer et.al, “Knowledge distillation: A good teacher is patient and consistent.” ArXiv, 2021
>
> [4]  Kolesnikov, “Big Transfer (BiT): General Visual Representation Learning”, ECCV, 2020

---

> > ### Comment · Reviewer_U2B6 · 2021-11-30
> > **Thank you for the response!**
> >
> > Thank you for addressing my concerns. I also agree with the other reviewers that this is a good paper, thus I increase my score.

---

### Official Review · Reviewer_aX8L · 2021-11-08

**Correctness:** 4
**Technical Novelty And Significance:** 3
**Empirical Novelty And Significance:** 3
**Recommendation:** 6
**Confidence:** 4

**Main Review:**



###Summary###

This paper enhances transferability of vision transformers (ViT) by introducing two strategies specific to the architecture of ViT models, i.e. Self-Ensemble and Token refinement. Specifically,  Self-Ensemble finds multiple discriminative pathways by dissecting a single ViT model into an ensemble of networks, leading to an explicit utilization of class-specific information at each ViT block. In addition, the proposed Token Refinement can potentially enhance the discriminative capacity at each block of ViT.

The high-level motivation of this paper is that the adversarial patterns found via conventional adversarial attacks show very low black-box transferability for ViT models. The authors claim that this phenomenon is due to the sub-optimal attack procedures that do not leverage the true representation potential of the ViTs. Thus, this paper introduces a highly transferable attack approach that augments the current adversarial attacks and increases their transferability from ViTs to the unknown models.

The paper conduct experiments on a range of standard attack methods to establish the performance boost obtained through the proposed transferablility approach by using the Source (white-box) models, Target (black-box) models. The experiments conducted on ImageNet/COCO dataset, and PASCAL demonstrate that the proposed method can outperform the baselines on some of the experimental settings. The paper also provides a detailed analysis of the model and experimental results.

### Novelty ###

The paper proposes two novel strategies about how to enhance the transferability of ViT models, i.e. refined tokens and self-ensemble. When those ideas are combined to an adversarial attack, it significantly boosts the transferability of adversarial examples, thereby bringing out the true generalization of ViTs’ adversarial space. These two ideas are interesting and should be heuristic for the research community, especially for those who have a strong interest in designing ViT models.


###Clarity###

The paper is overall logically clear and easy to follow. The tables and figures are well illustrated by the captions. The writing and presentation of this paper are good enough to convey the ideas.

###Pros###

1) The paper proposes two interesting ideas to enhance the generalization ability of the ViT models, i.e. refined tokens and self-ensemble. These two methods are well-motivated. For self-ensemble, it creates an ensemble of classifiers by learning a shared classification head at each block along the ViT hierarchy, leading to an explicit utilization of class-specific information at each ViT block. For refined tokens, the paper proposes both patch token refinement and class token refinement to tackle the misalignment between the features with the classifier.

2) The paper provides extensive experiments on multiple datasets, settings, and vision tasks. The results show that both proposed techniques can boost the performance, compared with many SOTA baselines.  Through the experiments, this paper empirically demonstrates favorable transfer rates across different model families (convolutional and transformer) as well as different vision tasks (classification, detection, and segmentation).

3) The paper provides a detailed empirical analysis to demonstrate the effectiveness of the proposed methods.

###Cons###

1) While self-ensembling is useful to enhance the transferability of the ViT models, it also makes the models more complex. Thus, some computational costs and inference speed analysis are desired.

2) One of the main assumptions of proposing the patch token refinement and class token refinement is there exists misalignment between the feature and classifier in ViT models. It will be helpful to provide some empirical evidence to validate this assumption (even some material in the appendix).

Based on the summary, cons, and pros, the current rating I am giving now is "weak accept". I would like to discuss the final rating with other reviewers, ACs.


**Summary Of The Paper:**


This paper enhances transferability of vision transformers (ViT) by introducing two strategies specific to the architecture of ViT models, i.e. Self-Ensemble and Token refinement. Specifically,  Self-Ensemble finds multiple discriminative pathways by dissecting a single ViT model into an ensemble of networks, leading to an explicit utilization of class-specific information at each ViT block. In addition, the proposed Token Refinement can potentially enhance the discriminative capacity at each block of ViT.

The high-level motivation of this paper is that the adversarial patterns found via conventional adversarial attacks show very low black-box transferability for ViT models. The authors claim that this phenomenon is due to the sub-optimal attack procedures that do not leverage the true representation potential of the ViTs. Thus, this paper introduces a highly transferable attack approach that augments the current adversarial attacks and increases their transferability from ViTs to the unknown models.

The paper conduct experiments on a range of standard attack methods to establish the performance boost obtained through the proposed transferablility approach by using the Source (white-box) models, Target (black-box) models. The experiments conducted on ImageNet/COCO dataset, and PASCAL demonstrate that the proposed method can outperform the baselines on some of the experimental settings. The paper also provides a detailed analysis of the model and experimental results.


**Summary Of The Review:**

Thia paper proposes two strategies specific to the architecture of ViT models, i.e. Self-Ensemble and Token refinement to enhance the transferability of the ViT models.  The experiments conducted on ImageNet/COCO dataset, and PASCAL demonstrate that the proposed method can outperform the baselines on some of the experimental settings.

---

> ### Author Response · Authors · 2021-11-22
> **Response to aX8L**
>
> We thank the reviewer for the encouraging and insightful comments. Please find our responses to specific queries below.
>
> **Computational Cost and Inference Speed Analysis:**
> As recommended, we analyze the computational cost along with the parameter complexity of our proposed approach in Appendix H. The basic version of our self-ensemble (Attack$^{E}$) only uses an off-the-shelf pretrained ViT without adding any additional parameters.  However, we introduce additional learnable parameters with our token refinement module to solve the misalignment problem between tokens produced by the intermediate blocks and the final classifier.   Our refinement module processes class and patch tokens using MLP (linear layer) and Convolutional block (Fig. 8 in Appendix H) and its parameter complexity is dependent on the embedding dimension of these pretrained tokens. For example, DeiT-S produces tokens with embedding dimension of size $\mathbb{R}^{384}$ and our refinement module adds 1.47 million parameters to its sub-model within the self-ensemble trained with refined tokens.
>
> Please note that fine-tuning with the additional parameters provides notable gains in recognition accuracy.  Our approach increases top-1 (%) accuracy (averaged across self-ensemble) by 12.43, 15.21, and 16.70 for Deit-T, Deit-S, and DeiT-B, respectively. A similar trend holds for the convolutional vision transformer [1] and MLP-Mixer [2] as well (Fig. 10 in Appendix I).
>
> We further analyze the inference speed for different attacks in Table-14 (Appendix H). The computational cost difference between the attack on a conventional model and the attack on our self-ensemble (without refinement module) is only marginal. As expected, attacking self-ensemble with our refinement module is slightly more expensive,  however, the notable difference is only with very large models such as DeiT-B. In fact, the increase in computational cost is a function of the original complexity of the pretrained ViT model e.g., DeiT-B with an original parametric complexity of 86 million generates high-dimensional tokens $\mathbb{R}^{784}$, which leads to higher compute cost in our refinement module.
>
> Thank you for suggesting this analysis. We will include the discussion on this tradeoff in our final manuscript.
>
> **On Misalignment between Features and the Classifier:**
> As recommended,  we visualize the latent space of our refined tokens in Fig. 9 (Appendix H.1). We randomly selected 10 classes from the ImageNet validation set distributed across the entire dataset. We extracted class tokens with and without refinement from the intermediate blocks (5,6,7,8) of Deit-T, Deit-S, and Deit-B. Our refined tokens have lower intra-class variations i.e., feature representations of samples from the same class are clustered together. Further refined tokens have better inter-class separation than the original tokens. This indicates that refinement minimizes the misalignment between the final classifier and intermediate class tokens, which leads to more disentangled representations. Attacking such disentangled representations across self-ensemble allows us to find better adversarial direction that leads to more powerful attacks. Further, the increase in top-1 (%) accuracy of the refined tokens from the intermediate blocks also highlights the benefit of the refinement module in solving the misalignment issue (Fig. 5 in Sec. 3.2).
>
>
> **References**
>
> [1] Wu et.al “CvT: Introducing Convolutions to Vision Transformers.” ArXiv, 2021
>
> [2] Tolstikhin et.al “MLP-Mixer: An all-MLP Architecture for Vision.” ArXiv, 2021

---

### Author Response · Authors · 2021-11-22
**Thank you for the valuable comments**

We thank all the reviewers (aX8L, U2B6, NMLw, HKsh) for their detailed and positive feedback. All reviewers appreciate the novelty of the proposed framework (aX8L, U2B6, NMLw, HKsh). Specifically, the proposed approach is well-motivated (aX8L, U2B6), generic, and likely to have a broad impact (HKsh). The transferability experiments have been noted to be extensive (aX8L, U2B6, HKsh), and the reviewers also appreciate clear writing and presentation of the paper (aX8L, U2B6, NMLw).

Our codes and trained models will be publicly released.

---

### Decision · Program_Chairs · 2022-01-20

**Decision:**

Accept (Spotlight)

**Comment:**

In this paper, the authors enhance the adversarial transferability of vision transformers by introducing two novel strategies specific to the architecture of ViT models: Self-Ensemble and Token Refinement method. Comprehensive experiments on various models (including CNN's and ViT's variants) and tasks (classification, detection, and segmentation) successfully verify the effectiveness of the proposed method.

In general, the problem studied is relevant and important. The paper is well-written and well-motivated with empirical findings. The proposed two strategies are novel, simple to implement, and effective in practice. Following the author's response and discussion, the average score increases from 6 to 7.5, with most concerns well addressed. AC believes that the paper should be highlighted at the ICLR conference.